# CausalVerse: Benchmarking Causal Representation Learning with Configurable High-Fidelity Simulations

**Guangyi Chen**[*1 2]    **Yunlong Deng**[*2]    **Peiyuan Zhu**[*2]    **Yan Li**[*2]

**Yifan Shen**[2]    **Zijian Li**[12]    **Kun Zhang**[12]

## Abstract

Causal Representation Learning (CRL) aims to uncover the data-generating process and identify the underlying causal variables and relations, whose evaluation remains inherently challenging due to the requirement of known ground-truth causal variables and causal structure. Existing evaluations often rely on either simplistic synthetic datasets or downstream performance on real-world tasks, generally suffering a dilemma between realism and evaluative precision. In this paper, we introduce a new benchmark for CRL using high-fidelity simulated visual data that retains both realistic visual complexity and, more importantly, access to ground-truth causal generating processes. The dataset comprises around 200 thousand images and 3 million video frames across 24 sub-scenes in four domains: static image generation, dynamic physical simulations, robotic manipulations, and traffic situation analysis. These scenarios range from static to dynamic settings, simple to complex structures, and single to multi-agent interactions, offering a comprehensive testbed that hopefully bridges the gap between rigorous evaluation and real-world applicability. In addition, we provide flexible access to the underlying causal structures, allowing users to modify or configure them to align with the required assumptions in CRL, such as available domain labels, temporal dependencies, or intervention histories. Leveraging this benchmark, we evaluated representative CRL methods across diverse paradigms and offered empirical insights to assist practitioners and newcomers in choosing or extending appropriate CRL frameworks to properly address specific types of real problems that can benefit from the CRL perspective. Welcome to visit our: **Project page:** causal-verse.github.io, **Dataset:** huggingface.co/CausalVerse.

## 1 Introduction

Understanding the causal factors underlying high-dimensional unstructured data, like images or videos, is a central challenge in machine learning and scientific discovery. Causal Representation Learning (CRL) has emerged as a promising framework to tackle this problem by recovering the latent causal variables and their relations from raw observations, or pursuing meaningful abstractions of fine-grained micro-variables. In general, CRL seeks to go beyond mere correlational patterns, offering representations that reflect the true generative mechanisms of data.

Despite rapid theoretical and methodological progress, the evaluation of CRL methods remains an open challenge. A major bottleneck lies in the absence of a real-world database with accessible ground-truth causal structures. This limitation forces existing evaluation strategies into a trade-off between realism and evaluative rigor. On the one hand, some methods adopt simplistic synthetic

---

[*]Equal contribution    [1]Carnegie Mellon University    [2]Mohamed bin Zayed University of Artificial Intelligence    Correspondence Email: `<guangyichen1994@gmail.com>`.

39th Conference on Neural Information Processing Systems (NeurIPS 2025) Track on Datasets and Benchmarks..

environments, such as physical process [1–3] or 3D object generation [4, 5], which enable precise verification but fall short in realism. For instance, Yao et al. [2] uses a physical setup where balls are connected by invisible springs to test whether CRL methods can recover object identities and their latent interactions. On the other hand, some works attempt to evaluate CRL indirectly via performance on real-world downstream tasks, such as domain adaptation/generalization [6, 7] and visual reasoning [8]. However, these real datasets lack annotated causal variables or structures, making it difficult to rigorously assess whether a method truly recovers causal factors or merely captures task-relevant correlations. This trade-off between realism and evaluative precision significantly hampers the ability to systematically compare and advance CRL methods.

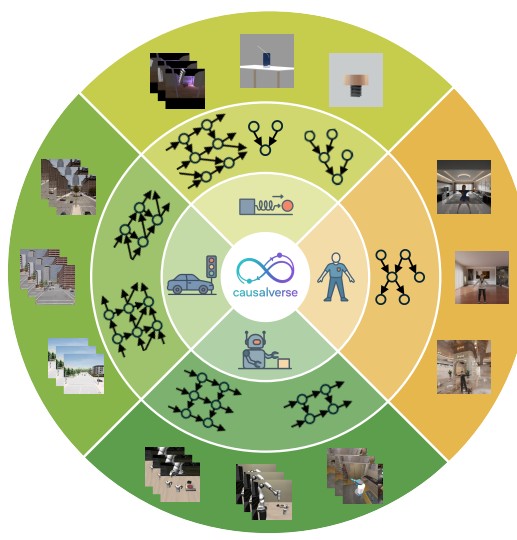

Figure 1: **The CausalVerse dataset.** Causal-Verse is organized with a hierarchical domain-scene-instance structure. We showcase four diverse domains: static generation, physical simulation, robotic manipulation, and traffic analysis. Within each domain, multiple scenes are designed to reflect distinct causal variables and structures. Sample instances are generated using simulation tools such as Unreal Engine 4, with each instance grounded in a predefined causal graph.

To address this limitation, we introduce Causal-Verse, a new benchmark specifically designed for causal representation learning. CausalVerse is a high-fidelity simulated visual dataset that combines realistic visual complexity with complete access to ground-truth causal variables and structures. Specifically, we carefully designed a set of scenes with well-defined causal factors and leveraged advanced rendering engines such as Blender and Unreal Engine 4 to generate high-quality simulated images and videos. The dataset includes around 200 thousand images and 300 million video frames across 24 sub-scenes drawn from four diverse domains: static image generation, dynamic physical simulations, robotic manipulation, and traffic scene analysis. These domains cover a broad spectrum of settings, ranging from static to dynamic, single-agent to multi-agent, and simple to highly structured environments. In addition, CausalVerse goes beyond static annotations by providing access to the entire simulation process, enabling configurable access to the underlying causal graphs, such as domain labels, temporal dependencies, and intervention histories. This flexibility allows researchers to modify the dataset to satisfy specific theoretical assumptions and experimental setups in CRL. As summarized in Table 1 compared to existing CRL benchmarks, CausalVerse offers greater scalability, richer diversity of task environments, high-dimensional latent spaces, and significantly improved realism, making it a more comprehensive and practical testbed for developing and evaluating causal representation methods.

Using this benchmark, we conduct a series of empirical evaluations of representative CRL methods grounded in diverse principles, providing comparative insights into their strengths, limitations, and applicability across a range of settings. Our results reveal that, despite recent theoretical advances in identifiability for CRL, applying these methods in practice remains challenging, particularly in scenarios involving complex visual content. Furthermore, we challenge existing CRL methods under unmet assumptions, offering practical guidance for researchers, especially newcomers who may not have prior knowledge of the specific assumptions underlying their data. By releasing CausalVerse along with these empirical analyses, we aim to advance the development of more robust and generalizable CRL methods, and support both researchers and practitioners in selecting appropriate tools for addressing real-world problems from a causal perspective.

We summarize the main contributions of this paper as follows: (1) **CausalVerse Benchmark:** A large-scale, high-fidelity visual dataset for CRL, featuring realistic complexity, high-dimensional latent spaces, and full access to ground-truth causal structures. (2) **Diverse Scenarios and Configurable Settings:** Around 200k images and 300 million video frames across 24 sub-scenes in four domains, with the whole simulation process to support configurable causal settings. (3) **Empirical Evaluation:**

Table 1: **Comparison of benchmarks related to Causal Representation Learning.** The number of video frames is used to represent the scale of each video dataset.

| Benchmark | Design for CRL | Data Type | Scale | Domain | Latent Variables | Dynamic Data | Ground Truth |
|---|---|---|---|---|---|---|---|
| Pendulum [29] | ✗ | Image | 7k | Physics | <10 | ✗ | ✓ |
| Flow [29] | ✗ | Image | 8k | Physics | <10 | ✗ | ✓ |
| CAUSAL3D [4] | ✗ | Image | 190k | Physics | <10 | ✗ | ✓ |
| Causal3DIdent [16] | ✓ | Image | 277k | 3D scenes | 10 | ✗ | ✓ |
| 3DIdent [5] | ✓ | Image | 275k | 3D scenes | 10 | ✗ | ✓ |
| CausalCircuit [18] | ✓ | Image | 120k | Electronics | 4 | ✗ | ✓ |
| Ball [3] | ✓ | Video | 2500k | Ball | <10 | ✓ | ✓ |
| Cloth [3] | ✓ | Video | 600k | Cloth | <10 | ✓ | ✓ |
| Light Tunnel [30] | ✓ | Image | 60k | Physical | 3-5 | ✓ | ✓ |
| ISTAnt [31] | ✗ | Video | 792k | Biology | - | ✓ | ✗ |
| CausalVerse | ✓ | Image Video | 198.66k 300m | Multi Domains | 3-129 | ✓ | ✓ |

Systematic comparison of CRL methods under varying assumptions and conditions. (4) **Practical Insights:** Guidance for selecting or designing CRL methods in real-world scenarios, especially under imperfect assumptions.

## 2  Related Work

**Causal representation learning.** CRL has emerged as a crucial paradigm in machine learning that aims to discover and model the underlying causal mechanisms that generate observable data [9]. While achieving identifiability by assuming the generating process is a linear mapping between latent variables and observations [10–14], extending the identifiability to nonlinear cases remains a significant challenge. Recently, different approaches are used to establish identifiability in nonlinear cases, like relying on sufficient changes in the latent variable distributions [1, 2, 6, 8, 15], supervised learning [16–18], multi-view [19, 20], and introducing structural constraints like sparsity [21–25].

**Evaluation by simple synthetics.** Simplified synthetic environments serve as common testbeds for CRL evaluation due to their precise ground-truth causal structures. Physical simulations represent a prominent approach, with works like V-CDN [3] and LEAP [2] utilizing mass-spring systems to assess whether methods can identify object identities and their latent interactions. Similarly, TDRL [1] evaluates temporal disentanglement in physical settings. Causal3DIdent [16], CAUSAL3D [4], and CausalCircuit [18] datasets offer images through 3D rendered engines like Blender [26] and MuJoCo [27], with controlled generative factors. But these images are about several simple observed objects and mechanisms, while also being limited to static and low-dimensional latent variables. Compared to existing datasets, CausalVerse presents more complex scenarios for causal representation learning across a wide range of scales—from static images to dynamic video, from single-agent to multi-agent interactions, and from simple low-dimensional variables to high-dimensional data with hundreds of features. It also leverages the more powerful Unreal Engine 4 [28] for rendering in some domains.

**Evaluation by downstream tasks.** Given the limitations of simplified simulations, researchers also evaluate CRL methods through performance on downstream tasks with real-world datasets, prioritizing practical utility while sacrificing precise causal verification. Transfer learning serves as a common task, with Sufficient Change [6] and SIG [32] assessing adaptation performance across domains to prove the usage of latent causal mechanisms, while Salaudeen et al. [7] analyze domain generalization datasets as proxy benchmarks for CRL. Reasoning [8, 33] and discovery task [34] provide another evaluation avenue for CRL. Image classification is also a widely used task for CRL methods [17, 19]. However, without ground-truth causal annotations, it remains unclear whether performance improvements stem from capturing genuine causal factors and mechanisms

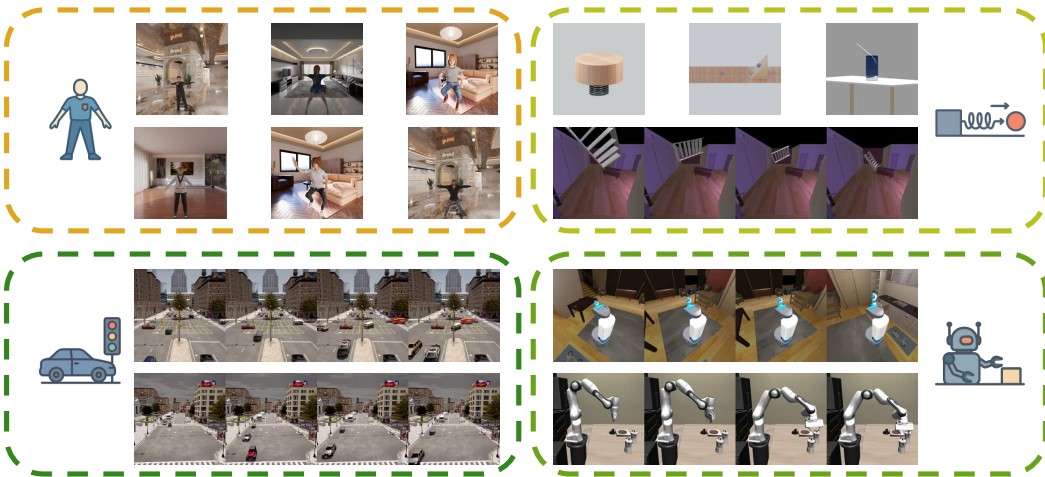

Figure 2: **Data examples of the dataset across four domains.** The Static Image Generation domain features the human images in diverse indoor environments. The Dynamic Physical Simulations domain describes different physical processes like spring compression, light refraction, and projectile motion. The Traffic Situation Analysis domain shows the traffic situations under different cities, scenes, and conditions. The Robotic Manipulations domain offers six scenes in which a robot operates in different indoor settings and performs various tasks.

or merely task-relevant correlations. CausalVerse integrates high-fidelity images and videos with comprehensive metadata, thereby supporting a broad spectrum of downstream tasks while enabling precise benchmarking against ground-truth latent variables and causal structures. In particular, its four static image generation scenarios can serve as distinct domains for domain adaptation experiments, and its robotics and physical simulation modules, each captured from multiple camera viewpoints, provide an exacting testbed for multi-view validation of causal models. More related work can be found in Appendix A.

## 3 The CausalVerse Dataset

In this section, we present CausalVerse, a high-fidelity simulated visual dataset specifically designed for CRL. While prior datasets often suffer from a trade-off between realism and access to ground-truth causal structures, CausalVerse is designed to bridge this gap by simulations offering both rich visual complexity and explicit, configurable causal ground truth. It supports rigorous evaluation of CRL methods under diverse and realistic conditions. To the best of our knowledge, CausalVerse is the most comprehensive and flexible benchmark for CRL to date. We introduce CausalVerse from the following perspectives: the composition of domains and sub-scenes, detailed dataset statistics, the simulation pipeline used to generate high-quality visual data and causal annotations, and finally, the flexibility it offers for modeling assumptions and real-world use cases.

### 3.1 Dataset composition

CausalVerse is designed to support a broad spectrum of CRL scenarios by offering diverse visual environments with clearly defined and accessible ground-truth causal structures. One of the core challenges in constructing such benchmarks is the difficulty of identifying or simulating scenarios where the true causal variables and relationships are known. As a result, most existing CRL datasets, even those using synthetic data, tend to focus narrowly on specific domains, such as 3D object generation [5] or simple physical systems [3, 4]. While these datasets are relatively easy to construct and evaluate, they often suffer from limited scenario diversity and poor scalability, which constrain their usefulness for testing CRL methods across a wide range of real-world conditions.

Towards the goal of constructing a large-scale benchmark with high diversity, the CausalVerse dataset is organized in a hierarchical domain–scene–instantiation structure, as illustrated in Figure 1. This design enables systematic variation across different levels of abstraction and supports comprehensive evaluation of CRL methods.

- **Domain.** For comprehensive evaluation, CausalVerse is organized into four distinct domains: static image generation, dynamic physical simulation, robotic manipulation, and traffic scene analysis, as showed in Figure 2. Each domain represents a unique category of visual and causal phenomena, ranging from object-centric generative processes in static settings to complex, multi-agent interactions in dynamic environments. This diversity enables the benchmark to capture a wide spectrum of causal challenges relevant to real-world scenarios.

- **Scene.** Within each domain, we construct multiple curated scenes (24 in total), each representing a specific causal setting. Scenes are designed to differ in key factors of the causal structure, such as the number and type of causal variables, the presence of domain-specific labels, and the visual context. This mid-level granularity enables researchers to evaluate CRL models under targeted structural assumptions and constraints. Taking the dynamic physical simulation domain as an example, CausalVerse includes 10 distinct scenes encompassing both aggregated and temporally dynamic cases. The aggregation cases condense physical processes into single images, capturing 4 phenomena such as a cylinder compressing a spring, light refraction, a ball decelerating and ascending a slope, and a ball freely falling onto plasticine. In contrast, the dynamic cases simulate continuous physical processes through videos, depicting falling, projectile motion, and collisions involving both single and interacting objects, with simple and complex motion mechanisms (6 scenes). The detailed data structure can be found in the Appendix.

- **Instance.** At the finest level, each scene includes a set of instantiations, concrete visual episodes generated by sampling from the underlying causal model. Each instantiation is accompanied by annotations of the relevant causal variables and their structural relationships. For example, in the traffic scene domain, altering the speed of a single vehicle can lead to the emergence of a traffic accident scenario, illustrating how specific changes in causal factors manifest in visually and semantically distinct outcomes. Such fine-grained control enables precise testing of a model's capacity to capture causal variables and relations across varying conditions.

A key strength of CausalVerse lies in its comprehensive coverage of diverse causal scenarios, making it uniquely positioned to evaluate CRL methods under a wide range of conditions. The dataset spans a spectrum of static to dynamic settings, including both image-based scenes with fixed causal factors and temporally evolving environments with sequential dependencies. It also covers structural complexity, from simple systems with a few causal variables to highly complex settings characterized by rich causal relations. Additionally, CausalVerse supports both single-agent and multi-agent interactions, enabling the study of causal reasoning in environments ranging from isolated object manipulation to collaborative or competitive agent behaviors, such as traffic systems or robotic coordination. This diversity ensures that models evaluated on CausalVerse are not only tested for technical soundness but also for their ability to generalize across realistic and varied causal contexts.

## 3.2 Dataset statistics

To illustrate the scale and diversity of the Causal-Verse dataset, we present key statistics summarizing its composition. In total, CausalVerse contains around 200k high-resolution images and around 140k videos with more than 300 million frames, distributed across 24 meticulously curated scenes spanning four distinct domains, including static image generation,

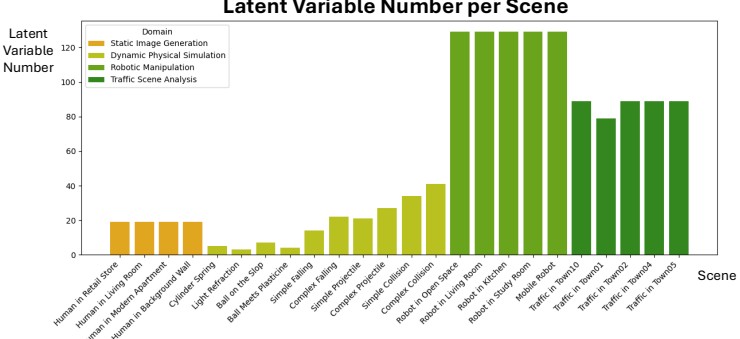

Figure 3: **Latent variable numbers across all scenes in CausalVerse.** please zoom in for details, like scene name.

dynamic physical simulation, robotic manipulation, and traffic scene analysis. As illustrated in Figure 4, these domains contain 4, 10, 5, and 5 scenes, respectively. The dynamic physical simulation domain is further subdivided into aggregated and dynamic categories, comprising 4 and 6 scenes, respectively. The detailed sample statistic of all senses is shown in the pie chart in Figure 4.

Each scene comprises hundreds to thousands of samples with video durations ranging from 3 to 32 seconds and a diverse set of frame rates. Frame resolutions vary across scenes, typically $1024 \times 1024$ or $1920 \times 1080$ pixels, thereby accommodating model training across different spatial scales and levels of visual detail. Each scene is governed by a set of causal variables, typically ranging from 3 to around 100, encompassing both categorical variables (e.g., object types, material categories) and continuous variables (e.g., velocity, mass, spatial coordinates). Figure 3 summarizes the distribution of the number of latent variables across all scenes. Here, the number of causal variables in the temporal setting refers to the variables present per

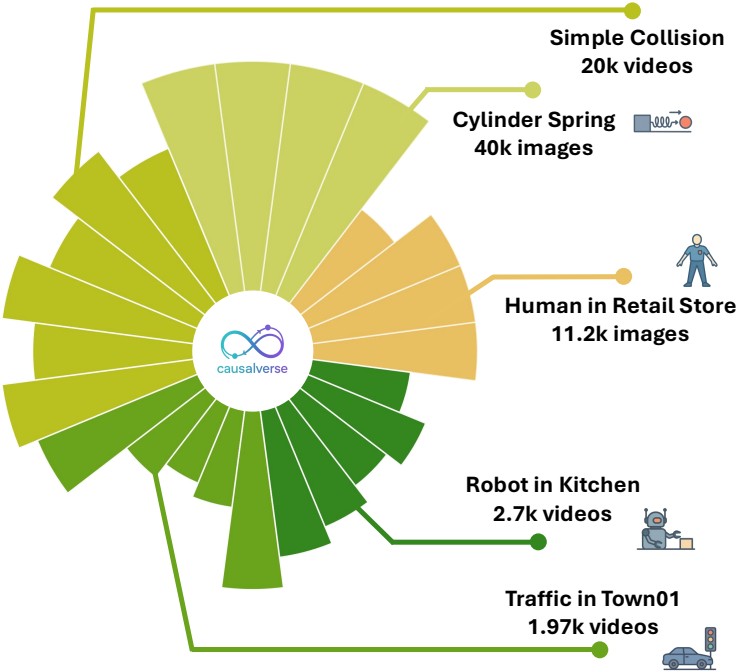

Figure 4: **Sample statistics across all scenes in CausalVerse.** The pie chart illustrates the relative data scale for all scenes, where the radius length reflects the number of images/videos (rather than frames), and colors indicate different domains. For example, the scene Cylinder Compressing Spring contains 40k images.

frame. In temporal processes, certain variables, such as object type or mass, remain invariant over time and serve as global descriptors of the system. In contrast, dynamic variables, including position, orientation, and momentum, continuously evolve throughout the video sequence, capturing the unfolding physical dynamics and enabling fine-grained causal reasoning over time. More dataset statistics can be found in Appendix B.

### 3.3 Data simulation pipeline

The data simulation process consists of three steps, including domain and scene definition, latent variable sampling, and image/video rendering. Figure 5 takes the domain of dynamic physical simulation as an example to illustrate the basic simulation process.

**Defining domains and scenes.** We generate data through a hierarchical progression approach. First, we select four distinct domains for our data generation framework: (1) static image generation, which showcases individuals with varying poses and appearances in different indoor environments; (2) physical simulations, including both complete videos documenting entire trajectories of objects in motion and time-lapse images capturing normalized physical processes where we record object positions at specific timestamps to create aggregated visualizations; (3) robotic manipulations; (4) traffic situation analysis. After determining the domain, we further define specific scenes. It is important to note that once a scene is established, the underlying causal variables and mechanisms become uniquely determined. We use metadata to document these hidden variables and mechanisms. For static image generation, scenes represent different indoor environments with distinct lighting, spatial dimensions, and object arrangements; for dynamic physical simulations, different scenes correspond to various physical processes and governing laws that determine object behaviors, robotic manipulations, and traffic conditions.

**Latent variable sampling.** Guided by the predefined causal graph for each scene, we sample latent variables based on their causal dependencies. For static scenes, we draw human-related factors (e.g., pose, attire, body shape) to form $D$-dimensional representations. For physical and robotic simulations, we sample a subset of physical variables (e.g., mass, position, rotation), with

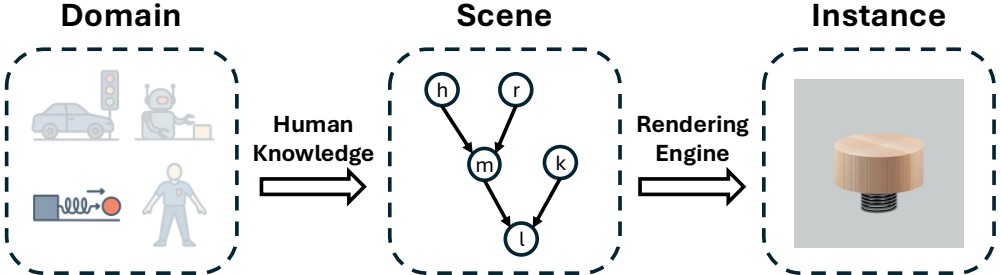

Figure 5: **Data construction pipeline illustrated using the dynamic physical simulation domain.** Starting from a task type such as dynamic physical simulation, we first identify candidate scenes and collect relevant causal variables and structures using human domain expertise. Instances are then generated using a rendering engine to ensure alignment with the designed causal structures.

others derived via physical laws to ensure temporal and physical consistency—resulting in $(T \times D)$-dimensional trajectories for dynamic settings. In traffic scenes, agent-level (e.g., vehicle position) and environment-level (e.g., weather) variables are sampled to capture structured interactions. These latent representations are then fed into domain-specific renderer, from neural engines to physics-based simulators, to generate high-fidelity outputs with causal coherence.

**Static rendering.** For static images, we uniformly employ the Blender engine to leverage its photorealistic rendering capabilities. Specifically, for human figures, we construct diverse character models utilizing the open-source assets for humans and clothing provided by the MakeHuman project [35]. For images depicting physical processes, we employ the Blenderproc [36] library, which automates simultaneous physical simulation and rendering.

**Dynamic rendering.** In generating dynamic videos, we select appropriate physics engines according to the characteristics of different domains. For dynamic physical simulations, we utilize the Bullet Physics SDK [37] and source assets from the Replica3D dataset [38]. For robotic simulation scenarios, we primarily build upon Robosuite [39] and Habitat-Lab, enabling embodied agents to navigate and interact within enclosed, highly interactive environments. For complex traffic scenarios, inspired by the Carla [40] project, we utilize Unreal Engine 4 to simulate diverse traffic conditions arising from variations in vehicle behaviors and traffic densities across different urban environments.

### 3.4 Flexibility and use cases

One of the key strengths of CausalVerse lies in its flexibility, which enables researchers to tailor the dataset to a wide range of CRL scenarios. By providing access to the full simulation process and underlying generative mechanisms, users can configure the dataset to align with specific theoretical assumptions or empirical needs.

We highlight two representative use cases:

- **Controlled experiments with satisfied assumptions.** In this setting, researchers can leverage the CausalVerse simulation pipeline to generate data that strictly adheres to key assumptions commonly required in CRL. These include, for example, the independent causal mechanism assumption, the availability of sufficient domain variation for identifiability, and specific functional priors such as sparsity, additivity, or linearity. By explicitly controlling the data-generating process, users can construct datasets that match the theoretical premises of CRL algorithms, allowing for precise and interpretable evaluation. This capability facilitates the empirical validation of identifiability results and learning guarantees, while still using visually rich, high-fidelity data that reflects the complexity of real-world observations.

- **Evaluation under unmet assumptions.** CausalVerse also supports the construction of more challenging and realistic scenarios where the data do not strictly adhere to standard theoretical assumptions, but instead reflect the complexity and ambiguity of real-world environments. Such settings may involve entangled causal factors, limited domain shifts, or ambiguous structural signals. These conditions are particularly valuable for stress-testing the robustness and generalization ability of CRL methods. They also offer practical insights for researchers, especially

newcomers, helping them better select or adapt CRL methods for real-world use, even if the precise knowledge of the data assumptions in their target applications is unknown.

This flexibility makes CausalVerse a powerful testbed not only for benchmarking but also for developing CRL methods that are both theoretically grounded and practically robust.

# 4 Evaluation

In this section, we first evaluate various CRL approaches that embody different principles under unmet assumptions in our scenarios, including both static images and temporal sequences. We also demonstrate how CausalVerse's flexibility allows us to organize data that satisfies the assumptions needed to test CRL methods. Through these experiments, we illustrate how CausalVerse enables researchers to thoroughly explore diverse CRL scenarios and approaches from empirical insights.

## 4.1 Evaluation metrics

To evaluate both component-wise and block-wise identifiability in CausalVerse, we adopt three metrics, including the Mean Correlation Coefficient (MCC), the coefficient of determination $R^2$, and the over-completed MCC. Specifically, let $Z \in \mathbb{R}^D$ be the ground-truth latent vector and $\widehat{Z} \in \mathbb{R}^{\widehat{D}}$ the estimated vector, we have:

- **Mean Correlation Coefficient (MCC):** To calculate MCC, we first compute the Pearson correlations $R_{ij} = \mathrm{corr}(Z_i, \widehat{Z}_j)$, then select an injective matching $\pi : \{1, \ldots, D\} \to \{1, \ldots, \widehat{D}\}$ maximizing $\sum_{i=1}^{D} |R_{i,\pi(i)}|$. Finally, the MCC value is defined as $\mathrm{MCC} = \frac{1}{D} \sum_{i=1}^{D} |R_{i,\pi(i)}|$.

- **Coefficient of Determination ($R^2$):** $R^2$ measures the proportion of variance in the ground-truth block $\mathbf{z}_b$ that is explained by the estimated block $\hat{\mathbf{z}}_b$. Formally,

$$R^2 = 1 - \frac{\mathrm{Var}(\mathbf{z}_b - f(\hat{\mathbf{z}}_b))}{\mathrm{Var}(\mathbf{z}_b)},$$

where $f$ is the regression function (linear or non-linear) that best predicts $\mathbf{z}_b$ from $\hat{\mathbf{z}}_b$. A value of $R^2 = 1$ indicates perfect block-wise identifiability.

- **Over-complete MCC:** In real-world applications, the true number of latent variables is typically unknown. Without model selection techniques, learned representations often become over-complete, containing both essential information and redundant or noisy components, i.e., $\widehat{D} > D$. To address this, we refine the MCC metric to focus only on the most informative components by selecting the top $D$ estimated variables that best match the ground-truth ones. We then apply the standard MCC computation over this subset to evaluate identifiability.

## 4.2 Implementation

To fairly compare the performance of different CRL methods on our dataset, we selected some representative ones to conduct unsupervised learning and estimate latent variables from images under unmet assumptions. These methods, based on established literature, include: Sufficient Change [6], which leverages the sufficient change principle to learn a generative model while incorporating structured assumptions from probabilistic graph modeling; Mechanism Sparsity [41], which utilizes sparsity constraints to encourage minimal dependencies between latent variables; Multiview [16], which learns the invariant representations by the alignment cross different views without explicit labels; and Contrastive Learning [42], which uses contrastive learning to identify the latent variables. Moreover, IDOL [24], CaRiNG [8], TDRL [1], TCL [43], and iVAE [44] are utilized for temporal causal representation learning from videos. All methods adopt the same VAE architecture in their implementations to ensure a fair comparison. Additionally, we introduced a supervised model, Supervised (encoder + MLP head trained on ground-truth latents), as an upper bound, incorporating an MLP layer after the ResNet output to learn latent variables with access to ground truth data.

## 4.3 Evaluation under unmet assumptions

To benchmark CRL methods under realistic scenarios, we directly train the models on our dataset, even when the data do not strictly satisfy the underlying assumptions of these methods. Specifically,

Table 2: **MCC and $R^2$ on image-based scenes under unsatisfied assumptions.**

| Algorithm | Ball on the Slope | | Cylinder Spring | | Light Refraction | | Avg | |
|---|---|---|---|---|---|---|---|---|
| | MCC | $R^2$ | MCC | $R^2$ | MCC | $R^2$ | MCC | $R^2$ |
| Supervised | 0.9878 | 0.9962 | 0.9970 | 0.9910 | 0.9900 | 0.9800 | 0.9916 | 0.9891 |
| Sufficient Change [6] | 0.4434 | 0.9630 | 0.6092 | 0.9344 | 0.6778 | 0.8420 | 0.5768 | 0.9131 |
| Mechanism Sparsity [41] | 0.2491 | 0.3242 | 0.3353 | 0.2340 | 0.1836 | 0.4067 | 0.2560 | 0.3216 |
| Multiview [16] | 0.4109 | 0.9658 | 0.4523 | 0.7841 | 0.3363 | 0.7841 | 0.3998 | 0.8447 |
| Contrastive Learning [42] | 0.2853 | 0.9604 | 0.6342 | 0.9920 | 0.3773 | 0.9677 | 0.4323 | 0.9734 |

we conduct experiments on three static scenes: Ball on the Slope, Cylinder Spring, and Light Refraction. Table 2 reports both the MCC and block-wise $R^2$ scores.

**Justification for Dissatisfaction.** The dataset is simulated to resemble realistic scenarios without imposing explicit constraints on the generation process. Thus, the assumptions for specific methods may not be satisfied. For example, Sufficient Change [6] requires enough $(2 * n_z + 1)$ domains to provide sufficient distribution changing, while our datasets only contain 4 views. Besides, we don't constrain the causal graph to be sparse, which conflicts with the assumptions in Mechanism Sparsity [41]. The Multiview [16] and Contrastive Learning [42] are designed for block-wise identification rather than the component-wise mapping. Nevertheless, we still evaluate MCC for these two methods to provide researchers with additional insights into their behavior.

**Results and Discussions.** Across the three scenes, the supervised upper bound achieves near-perfect performance on both metrics, confirming the encoder's capacity. However, current CRL methods remain unsatisfactory in terms of MCC, with all methods averaging below 0.6. This suggests a substantial gap still exists when applying current CRL approaches in real-world applications to obtain component-wise identifiable representations. Among current methods, Sufficient Change [6] gains the highest average MCC because this method is designed for component-wise identifiability, and it pays more attention to balancing reconstruction and invariance. For $R^2$, Sufficient Change [6], Multiview [16], and Contrastive Learning [42] all perform well, as the assumptions for block-wise identification are satisfied.

### 4.4 Evaluation for temporal causal representation learning

We evaluated five temporal CRL methods on two video scenes, Fall Simple and Robotics Study , representing the domains of Dynamic Physical Simulation and Robotic Manipulation, respectively. As shown in Table 3, temporal CRL under difficult dynamic scenarios remains challenging, as all methods exhibit low MCC values. Despite low absolute MCC values, we find that

Table 3: **MCC and $R^2$ on video-based scenes.**

| Algorithm | Fall Simple | | Robotics Study | |
|---|---|---|---|---|
| | MCC | $R^2$ | MCC | $R^2$ |
| IDOL [24] | 0.2527 | 0.5901 | 0.2500 | 0.6503 |
| CaRiNG [8] | 0.2280 | 0.5457 | 0.2225 | 0.6476 |
| TDRL [1] | 0.2003 | 0.5525 | 0.2440 | 0.6394 |
| TCL [43] | 0.1717 | 0.4892 | 0.2163 | 0.6150 |
| iVAE [44] | 0.1881 | 0.5233 | 0.1948 | 0.6165 |

methods incorporating sparsity constraints [24] and leveraging temporal context [8] achieve relatively better performance. Besides, we find that the Robotics Study scene, which contains denser relational structures, consistently yields higher $R^2$ values than the Fall Simple scene. Upon examining the data, we attribute this to the smaller object sizes in Fall Simple, which make the temporal relations more difficult to capture.

### 4.5 Evaluation for testing assumptions

We stress-test robustness by weakening the environment signal and corrupting domain labels in static image generation. We compare **Four-Scenes** (informative domains), **One-Scene** (insufficient variation), and **Wrong Scene Labels** (incorrect environmental indices). We report MCC and block-wise $R^2$ in Table 4. We find that reducing the number of domains weakens the sufficient-change signal and slightly degrades the performance of the Sufficient Change method. In contrast, injecting incorrect domain labels is even more detrimental, sometimes resulting in large negative $R^2$ values. Meanwhile,

Table 4: **Ablation on domain assumption violations in static image generation.**

| Algorithm | Four-Scenes | | One-Scene | | Wrong Scene Labels | | Avg | |
|---|---|---|---|---|---|---|---|---|
| | MCC | $R^2$ | MCC | $R^2$ | MCC | $R^2$ | MCC | $R^2$ |
| Supervised | 0.9001 | 0.6882 | 0.8646 | 0.5808 | 0.9001 | 0.6882 | 0.8883 | 0.6524 |
| Sufficient Change [6] | 0.3264 | 0.2898 | 0.3197 | 0.2671 | 0.1412 | $-6.7706$ | 0.2624 | $-2.0712$ |

the supervised upper bound remains stable across all settings, highlighting the requirement of sufficient change in distribution.

### 4.6 Configurable evaluation under specific assumptions

By open-sourcing the simulator with flexible control over the latent generation process, CausalVerse enables the evaluation of CRL methods under specific assumptions. For example, Sufficient Change decomposes the latent vector as $z = (z_c, z_s)$: $z_c$ is the content/invariant component and, in CausalVerse, aligns with the ground-truth latent that is shared across views; $z_s$ captures view-dependent variation. A key assumption of Sufficient Change is that at least $2n_s + 1$ distinct views are available, where $n_s$ is the dimensionality of $z_s$. For the configurable evaluation, we generate a new Cylinder Spring scene comprising 40040 images with enough views for testing Sufficient Change. As shown in Figure 6, adding enough domains shows a clear MCC improvement, which highlights the importance of suffi-

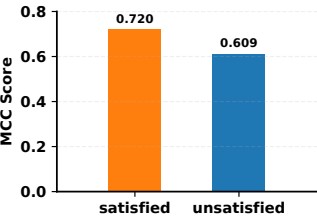

Figure 6: MCC comparison of sufficient change in satisfied and unmet dataset.

cient change assumptions. We expect that this flexible evaluation framework will enable researchers to adapt CausalVerse to align with the specific assumptions required by their methods.

## 5 Ethical Considerations, Limitations, and Conclusion

**Ethical considerations and limitations** CausalVerse is a simulated dataset generated using tools like Blender and Unreal Engine 4, containing no real-world or personal data, thus avoiding privacy or consent concerns. While care was taken to avoid biased or anthropomorphized representations, users should note that models trained on synthetic data may not generalize directly to real-world settings. Additionally, CausalVerse's causal structures are handcrafted and may not capture the full complexity or ambiguity of real-world systems. Despite our efforts to make the simulations as realistic as possible, there is still a gap between simulated data and realistic scenarios. The dataset also lacks natural noise factors such as sensor variability, and its visual complexity remains bounded by simulation capabilities. While full realism cannot yet be achieved, we are committed to moving in that direction by providing the research community with datasets that are as realistic, diverse, and large-scale as possible to support continued progress in CRL. While current rendering technologies still have limitations, we are fortunate to be at a time of rapid advancement in simulation, physically based rendering, and AI-generated content. These developments offer new opportunities to push the boundaries of synthetic realism.

**Conclusion** We present CausalVerse, a large-scale, high-fidelity benchmark for causal representation learning that aims to reconcile both realism with controllability and ground-truth access. By spanning a wide spectrum of domains, from static image generation to multi-agent traffic interactions, CausalVerse enables researchers to evaluate CRL methods across a broad range of diverse and challenging conditions. With fine-grained access to the underlying causal structures and simulation parameters, it further supports both the principled evaluation under idealized assumptions and practical stress testing in complex, realistic settings. Through empirical evaluation of representative approaches, we provide insights into the current state of CRL and its sensitivity to both satisfied and violated assumptions. We hope that CausalVerse serves as a stepping stone toward more reliable, interpretable, and generalizable causal learning systems, and as a foundation for future work that unites causal theory with complex visual environments.

## Acknowledgment

We would like to acknowledge the support from NSF Award No. 2229881, AI Institute for Societal Decision Making (AI-SDM), the National Institutes of Health (NIH) under Contract R01HL159805, and grants from Quris AI, Florin Court Capital, and MBZUAI-WIS Joint Program, and the Al Deira Causal Education project.

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

# NeurIPS Paper Checklist

1. **Claims**

   Question: Do the main claims made in the abstract and introduction accurately reflect the paper's contributions and scope?

   Answer: [Yes]

   Justification: We claim it at the end of Section 1

   Guidelines:

   - The answer NA means that the abstract and introduction do not include the claims made in the paper.
   - The abstract and/or introduction should clearly state the claims made, including the contributions made in the paper and important assumptions and limitations. A No or NA answer to this question will not be perceived well by the reviewers.
   - The claims made should match theoretical and experimental results, and reflect how much the results can be expected to generalize to other settings.
   - It is fine to include aspirational goals as motivation as long as it is clear that these goals are not attained by the paper.

2. **Limitations**

   Question: Does the paper discuss the limitations of the work performed by the authors?

   Answer: [Yes]

   Justification: The limitations are well discussed at Section 5.

   Guidelines:

   - The answer NA means that the paper has no limitation while the answer No means that the paper has limitations, but those are not discussed in the paper.
   - The authors are encouraged to create a separate "Limitations" section in their paper.
   - The paper should point out any strong assumptions and how robust the results are to violations of these assumptions (e.g., independence assumptions, noiseless settings, model well-specification, asymptotic approximations only holding locally). The authors should reflect on how these assumptions might be violated in practice and what the implications would be.
   - The authors should reflect on the scope of the claims made, e.g., if the approach was only tested on a few datasets or with a few runs. In general, empirical results often depend on implicit assumptions, which should be articulated.
   - The authors should reflect on the factors that influence the performance of the approach. For example, a facial recognition algorithm may perform poorly when image resolution is low or images are taken in low lighting. Or a speech-to-text system might not be used reliably to provide closed captions for online lectures because it fails to handle technical jargon.
   - The authors should discuss the computational efficiency of the proposed algorithms and how they scale with dataset size.
   - If applicable, the authors should discuss possible limitations of their approach to address problems of privacy and fairness.
   - While the authors might fear that complete honesty about limitations might be used by reviewers as grounds for rejection, a worse outcome might be that reviewers discover limitations that aren't acknowledged in the paper. The authors should use their best judgment and recognize that individual actions in favor of transparency play an important role in developing norms that preserve the integrity of the community. Reviewers will be specifically instructed to not penalize honesty concerning limitations.

3. **Theory assumptions and proofs**

   Question: For each theoretical result, does the paper provide the full set of assumptions and a complete (and correct) proof?

   Answer: [NA]

Justification: This paper is only about dataset and benchmark.

Guidelines:

- The answer NA means that the paper does not include theoretical results.
- All the theorems, formulas, and proofs in the paper should be numbered and cross-referenced.
- All assumptions should be clearly stated or referenced in the statement of any theorems.
- The proofs can either appear in the main paper or the supplemental material, but if they appear in the supplemental material, the authors are encouraged to provide a short proof sketch to provide intuition.
- Inversely, any informal proof provided in the core of the paper should be complemented by formal proofs provided in appendix or supplemental material.
- Theorems and Lemmas that the proof relies upon should be properly referenced.

4. **Experimental result reproducibility**

Question: Does the paper fully disclose all the information needed to reproduce the main experimental results of the paper to the extent that it affects the main claims and/or conclusions of the paper (regardless of whether the code and data are provided or not)?

Answer: [Yes]

Justification: The code is in the supplementary material and experiment detail can be found in Appendix.

Guidelines:

- The answer NA means that the paper does not include experiments.
- If the paper includes experiments, a No answer to this question will not be perceived well by the reviewers: Making the paper reproducible is important, regardless of whether the code and data are provided or not.
- If the contribution is a dataset and/or model, the authors should describe the steps taken to make their results reproducible or verifiable.
- Depending on the contribution, reproducibility can be accomplished in various ways. For example, if the contribution is a novel architecture, describing the architecture fully might suffice, or if the contribution is a specific model and empirical evaluation, it may be necessary to either make it possible for others to replicate the model with the same dataset, or provide access to the model. In general. releasing code and data is often one good way to accomplish this, but reproducibility can also be provided via detailed instructions for how to replicate the results, access to a hosted model (e.g., in the case of a large language model), releasing of a model checkpoint, or other means that are appropriate to the research performed.
- While NeurIPS does not require releasing code, the conference does require all submissions to provide some reasonable avenue for reproducibility, which may depend on the nature of the contribution. For example
  (a) If the contribution is primarily a new algorithm, the paper should make it clear how to reproduce that algorithm.
  (b) If the contribution is primarily a new model architecture, the paper should describe the architecture clearly and fully.
  (c) If the contribution is a new model (e.g., a large language model), then there should either be a way to access this model for reproducing the results or a way to reproduce the model (e.g., with an open-source dataset or instructions for how to construct the dataset).
  (d) We recognize that reproducibility may be tricky in some cases, in which case authors are welcome to describe the particular way they provide for reproducibility. In the case of closed-source models, it may be that access to the model is limited in some way (e.g., to registered users), but it should be possible for other researchers to have some path to reproducing or verifying the results.

5. **Open access to data and code**

Question: Does the paper provide open access to the data and code, with sufficient instructions to faithfully reproduce the main experimental results, as described in supplemental material?

Answer: [Yes]

Justification:The code is in the supplementary material.

Guidelines:

- The answer NA means that paper does not include experiments requiring code.
- Please see the NeurIPS code and data submission guidelines (`https://nips.cc/public/guides/CodeSubmissionPolicy`) for more details.
- While we encourage the release of code and data, we understand that this might not be possible, so "No" is an acceptable answer. Papers cannot be rejected simply for not including code, unless this is central to the contribution (e.g., for a new open-source benchmark).
- The instructions should contain the exact command and environment needed to run to reproduce the results. See the NeurIPS code and data submission guidelines (`https://nips.cc/public/guides/CodeSubmissionPolicy`) for more details.
- The authors should provide instructions on data access and preparation, including how to access the raw data, preprocessed data, intermediate data, and generated data, etc.
- The authors should provide scripts to reproduce all experimental results for the new proposed method and baselines. If only a subset of experiments are reproducible, they should state which ones are omitted from the script and why.
- At submission time, to preserve anonymity, the authors should release anonymized versions (if applicable).
- Providing as much information as possible in supplemental material (appended to the paper) is recommended, but including URLs to data and code is permitted.

6. **Experimental setting/details**

Question: Does the paper specify all the training and test details (e.g., data splits, hyper-parameters, how they were chosen, type of optimizer, etc.) necessary to understand the results?

Answer: [Yes]

Justification: Experiment settings and methods can be found in Section 4 and more details can be found in the Appendix.

Guidelines:

- The answer NA means that the paper does not include experiments.
- The experimental setting should be presented in the core of the paper to a level of detail that is necessary to appreciate the results and make sense of them.
- The full details can be provided either with the code, in appendix, or as supplemental material.

7. **Experiment statistical significance**

Question: Does the paper report error bars suitably and correctly defined or other appropriate information about the statistical significance of the experiments?

Answer: [Yes]

Justification:The Standard Deviation are concluded in Appendix due to limit of main text.

Guidelines:

- The answer NA means that the paper does not include experiments.
- The authors should answer "Yes" if the results are accompanied by error bars, confidence intervals, or statistical significance tests, at least for the experiments that support the main claims of the paper.
- The factors of variability that the error bars are capturing should be clearly stated (for example, train/test split, initialization, random drawing of some parameter, or overall run with given experimental conditions).
- The method for calculating the error bars should be explained (closed form formula, call to a library function, bootstrap, etc.)
- The assumptions made should be given (e.g., Normally distributed errors).

- It should be clear whether the error bar is the standard deviation or the standard error of the mean.
- It is OK to report 1-sigma error bars, but one should state it. The authors should preferably report a 2-sigma error bar than state that they have a 96% CI, if the hypothesis of Normality of errors is not verified.
- For asymmetric distributions, the authors should be careful not to show in tables or figures symmetric error bars that would yield results that are out of range (e.g. negative error rates).
- If error bars are reported in tables or plots, The authors should explain in the text how they were calculated and reference the corresponding figures or tables in the text.

8. **Experiments compute resources**

   Question: For each experiment, does the paper provide sufficient information on the computer resources (type of compute workers, memory, time of execution) needed to reproduce the experiments?

   Answer: [Yes]

   Justification: We reported it in Appendix.

   Guidelines:
   - The answer NA means that the paper does not include experiments.
   - The paper should indicate the type of compute workers CPU or GPU, internal cluster, or cloud provider, including relevant memory and storage.
   - The paper should provide the amount of compute required for each of the individual experimental runs as well as estimate the total compute.
   - The paper should disclose whether the full research project required more compute than the experiments reported in the paper (e.g., preliminary or failed experiments that didn't make it into the paper).

9. **Code of ethics**

   Question: Does the research conducted in the paper conform, in every respect, with the NeurIPS Code of Ethics https://neurips.cc/public/EthicsGuidelines?

   Answer: [Yes]

   Justification: We strictly follow the Code of Ethics.

   Guidelines:
   - The answer NA means that the authors have not reviewed the NeurIPS Code of Ethics.
   - If the authors answer No, they should explain the special circumstances that require a deviation from the Code of Ethics.
   - The authors should make sure to preserve anonymity (e.g., if there is a special consideration due to laws or regulations in their jurisdiction).

10. **Broader impacts**

    Question: Does the paper discuss both potential positive societal impacts and negative societal impacts of the work performed?

    Answer: [Yes]

    Justification: We have a statement in Appendix.

    Guidelines:
    - The answer NA means that there is no societal impact of the work performed.
    - If the authors answer NA or No, they should explain why their work has no societal impact or why the paper does not address societal impact.
    - Examples of negative societal impacts include potential malicious or unintended uses (e.g., disinformation, generating fake profiles, surveillance), fairness considerations (e.g., deployment of technologies that could make decisions that unfairly impact specific groups), privacy considerations, and security considerations.

- The conference expects that many papers will be foundational research and not tied to particular applications, let alone deployments. However, if there is a direct path to any negative applications, the authors should point it out. For example, it is legitimate to point out that an improvement in the quality of generative models could be used to generate deepfakes for disinformation. On the other hand, it is not needed to point out that a generic algorithm for optimizing neural networks could enable people to train models that generate Deepfakes faster.
- The authors should consider possible harms that could arise when the technology is being used as intended and functioning correctly, harms that could arise when the technology is being used as intended but gives incorrect results, and harms following from (intentional or unintentional) misuse of the technology.
- If there are negative societal impacts, the authors could also discuss possible mitigation strategies (e.g., gated release of models, providing defenses in addition to attacks, mechanisms for monitoring misuse, mechanisms to monitor how a system learns from feedback over time, improving the efficiency and accessibility of ML).

11. **Safeguards**

Question: Does the paper describe safeguards that have been put in place for responsible release of data or models that have a high risk for misuse (e.g., pretrained language models, image generators, or scraped datasets)?

Answer: [NA]

Justification: The paper has no such risk.

Guidelines:

- The answer NA means that the paper poses no such risks.
- Released models that have a high risk for misuse or dual-use should be released with necessary safeguards to allow for controlled use of the model, for example by requiring that users adhere to usage guidelines or restrictions to access the model or implementing safety filters.
- Datasets that have been scraped from the Internet could pose safety risks. The authors should describe how they avoided releasing unsafe images.
- We recognize that providing effective safeguards is challenging, and many papers do not require this, but we encourage authors to take this into account and make a best faith effort.

12. **Licenses for existing assets**

Question: Are the creators or original owners of assets (e.g., code, data, models), used in the paper, properly credited and are the license and terms of use explicitly mentioned and properly respected?

Answer: [Yes]

Justification: The baselines and datasets are used and cited correctly.

Guidelines:

- The answer NA means that the paper does not use existing assets.
- The authors should cite the original paper that produced the code package or dataset.
- The authors should state which version of the asset is used and, if possible, include a URL.
- The name of the license (e.g., CC-BY 4.0) should be included for each asset.
- For scraped data from a particular source (e.g., website), the copyright and terms of service of that source should be provided.
- If assets are released, the license, copyright information, and terms of use in the package should be provided. For popular datasets, `paperswithcode.com/datasets` has curated licenses for some datasets. Their licensing guide can help determine the license of a dataset.
- For existing datasets that are re-packaged, both the original license and the license of the derived asset (if it has changed) should be provided.

- If this information is not available online, the authors are encouraged to reach out to the asset's creators.

13. **New assets**

    Question: Are new assets introduced in the paper well documented and is the documentation provided alongside the assets?

    Answer: [Yes]

    Justification: We have a new benchmark and produced a new dataset, all of these are released in Hugging Face, the link can be found in the abstract.

    Guidelines:

    - The answer NA means that the paper does not release new assets.
    - Researchers should communicate the details of the dataset/code/model as part of their submissions via structured templates. This includes details about training, license, limitations, etc.
    - The paper should discuss whether and how consent was obtained from people whose asset is used.
    - At submission time, remember to anonymize your assets (if applicable). You can either create an anonymized URL or include an anonymized zip file.

14. **Crowdsourcing and research with human subjects**

    Question: For crowdsourcing experiments and research with human subjects, does the paper include the full text of instructions given to participants and screenshots, if applicable, as well as details about compensation (if any)?

    Answer: [NA]

    Justification: This paper does not involve crowdsourcing nor research with human subjects.

    Guidelines:

    - The answer NA means that the paper does not involve crowdsourcing nor research with human subjects.
    - Including this information in the supplemental material is fine, but if the main contribution of the paper involves human subjects, then as much detail as possible should be included in the main paper.
    - According to the NeurIPS Code of Ethics, workers involved in data collection, curation, or other labor should be paid at least the minimum wage in the country of the data collector.

15. **Institutional review board (IRB) approvals or equivalent for research with human subjects**

    Question: Does the paper describe potential risks incurred by study participants, whether such risks were disclosed to the subjects, and whether Institutional Review Board (IRB) approvals (or an equivalent approval/review based on the requirements of your country or institution) were obtained?

    Answer: [NA]

    Justification: This paper does not involve crowdsourcing nor research with human subjects

    Guidelines:

    - The answer NA means that the paper does not involve crowdsourcing nor research with human subjects.
    - Depending on the country in which research is conducted, IRB approval (or equivalent) may be required for any human subjects research. If you obtained IRB approval, you should clearly state this in the paper.
    - We recognize that the procedures for this may vary significantly between institutions and locations, and we expect authors to adhere to the NeurIPS Code of Ethics and the guidelines for their institution.
    - For initial submissions, do not include any information that would break anonymity (if applicable), such as the institution conducting the review.

16. **Declaration of LLM usage**

    Question: Does the paper describe the usage of LLMs if it is an important, original, or non-standard component of the core methods in this research? Note that if the LLM is used only for writing, editing, or formatting purposes and does not impact the core methodology, scientific rigorousness, or originality of the research, declaration is not required.

    Answer: [NA]

    Justification: We don't invole llm in core method development.

    Guidelines:

    - The answer NA means that the core method development in this research does not involve LLMs as any important, original, or non-standard components.
    - Please refer to our LLM policy (`https://neurips.cc/Conferences/2025/LLM`) for what should or should not be described.

*Appendix for*
# "CausalVerse: Benchmarking Causal Representation Learning with Configurable High-Fidelity Simulations"

## A    Detailed Related Work

**Causal discovery.**    Understanding complex systems lies in causal discovery, the identification of causal relations from observational data [45]. Causal discovery methods generally fall into three classes: constraint-based approaches use conditional-independence tests to recover the causal skeleton and orient edges up to a Markov equivalence class (e.g., PC [45] and its kernel-based KCI extension [46], FCI for latent confounding and selection bias [45, 47], and CCD for feedback loops [48]); score-based techniques frame structure learning as model selection over DAGs—most notably Greedy Equivalence Search (GES) [49], which maximizes a penalized likelihood (such as BIC [50]) and admits extensions for nonlinear relationships [51]; and functional causal model (FCM) methods impose specific assumptions on the data-generating process [52] (e.g., LiNGAM for linear non-Gaussian systems [53], nonlinear additive noise models [54, 55], and post-nonlinear models [56]) to secure identifiability of the underlying causal graph.

**Latent variables and constraints.**    Although the causal framework is well established, in many real-world scenarios the variables of interest are latent constructs that cannot be directly observed or quantified. A first try to handle latent variables for causal discovery is FCI [45], however, it is already maximally informative under nonparametric CI constraints [57, 47]. Therefore, many new tools beyond CI constraints have thus been developed, typically by imposing additional parametric assumptions. These include rank constraints [58, 14], which generalize the Tetrad representation theorem from [45] and provide algebraic conditions on covariance matrices; equality constraints derived from Gaussian structural equation models that even have rank constraints as a subclass [59], high-order moment constraints (beyond the second-order moments in statistics, e.g., skewness, kurtosis, etc.) [60, 61, 8], which exploit non-Gaussianity for identifiability. Additionally, constraints based on matrix decomposition [62], copula models [63], and mixture oracles [64] have also been developed.

**Causal representation learning.**    Causal Representation Learning (CRL) has emerged as a crucial paradigm in machine learning that aims to discover and model the underlying causal mechanisms that generate observable data [9]. While achieving identifiability by assuming the generating process is a linear mapping between latent variables and observations [10–14], extending identifiability to nonlinear cases remains a significant challenge. Recently, different approaches have been used to establish identifiability in nonlinear settings. One major direction relies on sufficient changes in latent variable distributions, where nonstationary or environmental shifts enable recovery of nonlinear independent components and causal structures [6, 15, 8, 1, 2]. Supervised learning approaches incorporate auxiliary information, self-supervised learning, or weak supervision to constrain the representation learning problem [16–18]. Multi-view learning exploits simultaneously observed data modalities to identify shared causal factors through contrastive learning frameworks [19, 20]. Additionally, structural constraints like sparsity regularize either the causal graph structure or the mixing mechanisms to achieve identifiability [21–25]. These diverse methodological directions reflect the inherent complexity of nonlinear identifiability and highlight the necessity of systematic benchmarking to evaluate and compare different approaches for recovering meaningful causal representations from high-dimensional observational data.

**Simulation approaches**    Contemporary CRL datasets emerge from four distinct simulation approaches that trace a realism–controllability spectrum. (i) At the low-fidelity extreme, Pendulum [29] and Flow [29] exemplify analytic toy-physics worlds whose closed-form ODEs and minimalist ray tracing enable millisecond image generation and pixel-perfect ground truth. (ii) Stepping up in complexity, mass–spring and rigid-body engines such as MuJoCo [27], PhysX and Box2D drive datasets like Ball [3] and Cloth [3], enriching dynamics with contact, friction, and deformable bodies. (iii) A different flavor appears in task-oriented robotics and circuit simulators: CausalCircuit [18] couples MuJoCo [27] robot arms with digital-logic models so that perception, action, and outcome are co-simulated within a single loop. (iv) Pushing for visual realism, high-fidelity 3D renderers such

as Blender Cycles, underpin 3DIdent [5], Causal3DIdent [16], and CAUSAL3D [4], where photoreal-istic materials, lighting, and multi-view cameras expand latent spaces from a handful of factors to dozens. Beyond Blender, CausalVerse employs simulation engines such as Unreal Engine 4 [28] to generate high-quality images and videos. For robotic interactions, the Habitat-Sim and RoboSuite platforms are used for simulation; these platforms are based on the Bullet and MuJoCo [27] physics engines, respectively.

**Evaluation by simple synthetics.** Simplified synthetic environments serve as common testbeds for CRL evaluation due to their precise ground-truth causal structures. Physical simulations represent a prominent approach, with works like V-CDN [3] and LEAP [2] utilizing mass-spring systems to assess whether methods can identify object identities and their latent interactions. Similarly, TDRL [1] evaluates temporal disentanglement in physical settings. The Causal3DIdent [16], CAUSAL3D [4], and CausalCircuit [18] datasets offer images through 3D rendering engines like Blender and MuJoCo, with controlled generative factors. However, these images depict only simple observed objects and mechanisms and are limited to static, low-dimensional latent variables. Compared to existing datasets, CausalVerse presents more complex scenarios for causal representation learning across a wide range of scales—from static images to dynamic video, from single-agent to multi-agent interactions, and from simple low-dimensional variables to high-dimensional data with hundreds of features. It also leverages the more powerful Unreal Engine 4 [28] for rendering in some domains.

**Evaluation by downstream tasks.** Given the limitations of simplified simulations, researchers also evaluate CRL methods through performance on downstream tasks with real-world datasets, prioritizing practical utility while sacrificing precise causal verification. Transfer learning serves as a common task, with iMSDA [6] and SIG [32] assessing adaptation performance across domains to demonstrate the usage of latent causal mechanisms, while Salaudeen et al. [7] analyze domain generalization datasets as proxy benchmarks for CRL. Reasoning [8, 33] and discovery tasks [34] provide another evaluation avenue for CRL. Image classification is also a widely used task for CRL methods [19, 17]. However, without ground-truth causal annotations, it remains unclear whether performance improvements stem from capturing genuine causal factors and mechanisms or merely task-relevant correlations. CausalVerse integrates high-fidelity images and videos with comprehensive metadata, thereby supporting a broad spectrum of downstream tasks while enabling precise benchmarking against ground-truth latent variables and causal structures. In particular, its four static image generation scenarios can serve as distinct domains for domain-adaptation experiments, and its robotics and physical-simulation modules—each captured from multiple camera viewpoints—provide an exacting testbed for multi-view validation of causal models.

# B Data Structure, Example Showcase, Brief Comparison and Discussions

In this section, we present an overview of the dataset's structural design, summarized in a comprehen-sive table. We then showcase detailed examples from various scenes to illustrate the dataset's diversity. Finally, we provide a brief comparison of the realism between our dataset and other existing synthetic datasets. As shown in Figure 2, our CausalVerse dataset is organized into 4 domains comprising 24 distinct scenes. In the sections that follow, we will introduce each scene's data scale, its causal graph, an accompanying description of the corresponding scenario, and an example showcase.

## B.1 Construction of causal relationships

In our dataset, these relationships arise from three sources: **(1) physical constraints, (2) basic social rules, and (3) arbitrary human constructs.** For physics-based processes and robotic domains, the edge set and functional dependencies are chosen to align with established physical laws, so that cause and effect follow well-known principles rather than ad hoc assumptions. For example, state updates and interactions are specified in a way that is consistent with standard mechanics, using conventional variables and units, and with directions of influence that reflect accepted domain knowledge. In the traffic domain, the causal influences encode social conventions such as maintaining safe following distances, avoiding collisions, and stopping at red lights. Here the intent is to represent plausible driver and agent behavior as shaped by widely understood rules, without claiming to capture every nuance of real world decision making. In the static image generation domain, the causal structures are deliberately constructed as human designs. They are intentionally built to support flexible data

manipulation and to make relations more complex, so that users can create controlled dependencies and diverse appearance variations for the purpose of studying causal representation learning. These graphs are not intended to represent realistic causality; rather, they serve as structured tools that organize factors and expose interpretable levers for generating images under different controlled settings. We explicitly acknowledge that these arbitrarily designed graphs apply only to the static image generation domain and do not reflect real world causal structures. For all graphs that are derived from physical laws or social rules, we take care to ensure plausibility and internal consistency during design. In practice, this means we define variables with clear meanings and admissible ranges, specify directions of influence that are consistent with accepted understanding, keep parameters within reasonable magnitudes, and avoid cycles or contradictions at the level of the stated mechanisms. We also check that the qualitative implications of the specified relationships are sensible under simple variations of conditions, so that the resulting graphs remain stable and coherent.

## B.2 Static image generation

The static image generation domain is fully parameterized by 19 controllable variables that jointly govern the synthesis and rendering of a human subject in a single frame. We further partition this domain into four distinct scenes, each corresponding to a unique indoor setting. Each setting is characterized by diverse lighting conditions and distinct background layouts. As summarized in Table B.1, the domain comprises four scene categories: Human in Retail Store, Human in Living Room, Human in Modern Apartment, and Human against Background Wall. The showcase examples can be found in Figures B.1 and B.2.

Table B.1: Causal graph, data size, and description for the domain static image generation

| Showcases | Data Size | Causal Graph | Description |
|---|---|---|---|
| Figures B.1 and B.2 | ~40 k |  | This figure presents a **partial causal graph** for static character generation and rendering. Variables shown are gender (gnd), cup size (cup), clothing style (clo), body weight (wgt), overall appearance (app), and eyebrows (ebr). Gender directly influences clothing style and cup size, and these two variables together shape the character's final appearance. Other factors, such as weight and eyebrows, act independently, each contributing to the rendered appearance without interacting with one another. |

## B.3 Dynamic physical simulations (aggregated image)

We provide four aggregated image-based dynamic physical scenarios in Table B.2, which are arranged to impose a controlled escalation of causal dimensionality, ranging from a three-factor system up to a seven-factor environment. Specifically, the Cylinder–Spring scene comprises five latent variables; the Light Refraction scenario involves three latent variables; the Ball on the Slope configuration encompasses seven latent variables; and the Ball Meets Plasticine sequence comprises four latent variables. In each case, every sample provides multiple visually inferable quantities—such as compression length, refracted angle, sliding distance, or penetration depth—from which one can, under the physical laws, deduce additional, less immediately apparent factors (e.g., spring stiffness, initial velocity, medium viscosity). Moreover, each sample includes multiple views of the same scene to facilitate diverse research tasks. The detailed image examples can be found in Figures B.3 to B.6.

## B.4 Dynamic physical simulations (video)

For the video-based dynamic physical simulations domain, we provide six distinct scenes: Fall Simple, Fall Complex, Projectile Simple, Projectile Complex, Collision Simple, and Collision Complex. Regarding the distinction between simple and complex scenes: for Fall, the simple

Table B.2: Causal graph, data size, and scene description for aggregated images for dynamic physical simulations

| Scene | Data size | Causal graph | Description |
|---|---|---|---|
| Cylinder Spring

Figure B.3 | 40k |  | This scene simulates a homogeneous cylinder compressing an ideal spring under gravity until static equilibrium, with the cylinder's mass scaling as $m \propto hr^2$ (where $h$ and $r$ denote the cylinder's height and radius, respectively) and the resulting spring deformation scaling as $l \propto m/k$, where $k$ is the spring stiffness coefficient. |
| Light Refraction

Figure B.4 | 40k |  | This scene simulates a collimated light ray refracting at a flat boundary between air and an aqueous ink solution of varying concentration (and hence varying refractive index $n_2$). For each sample, the incident angle $\theta_1$ and ink concentration are changed, which determines $n_2$, and the refracted angle $\theta_2$ is computed via Snell's law $\sin(\theta_1) = n_2 \sin(\theta_2)$. Images capture the resulting ray path for downstream causal representation evaluation. |
| Ball on the Slope

Figure B.5 | 40k |  | This scene simulates a small sphere given an initial speed $v_1$ across a horizontal surface for a fixed time $2T$. The table friction coefficient $\mu_1$ is proportional to the sphere's roughness $r$, causing it to decelerate to $v_2 = v_1 - \mu_1 g(2T)$. The sphere then ascends an incline of angle $\theta$ with slope friction $\mu_2 \propto r$, traveling a distance $l = \frac{v_2^2}{2g(\sin\theta + \mu_2 \cos\theta)}$. Each sample varies $v_1$, $\theta$, and roughness $r$ to generate diverse sliding distances $l$. |
| Ball Meets Plasticine

Figure B.6 | 40k |  | This scene simulates a small sphere whose mass $m \propto r^3$ undergoing free fall from a height $h$ under gravity $g$. Upon striking a viscous clay medium with viscosity coefficient $u$, the sphere penetrates to a depth $l = \frac{m}{u}\sqrt{2gh}$. Each sample varies $h$, $r$, and $u$, producing diverse penetration depths $l$. |

version contains only a single object undergoing free fall, without additional global factors such as lighting. For Projectile, the complex version introduces object-specific angular velocity, vertical linear velocity, and global lighting conditions. For Collision, the complex version adds distinct angular velocities to both objects and incorporates global lighting changes. Each scene features multiple camera viewpoints, which opens up opportunities for exploring view-specific features and their entanglements. We group the six scenes into three categories: Fall, Projectile, and Collision. A more intuitive and detailed overview of the representative simple scenes is provided in Table B.3. Since the complex scenes are essentially extensions of their simple counterparts, we selectively showcase

representative complex examples, while illustrative examples of the complex scenes can be found in Figures B.7 to B.9.

## B.5    Robotic manipulations

For the Robotic Manipulations domain, we provide five distinct scenes: Kitchen, Living, Study, General, and Mobile. The first four involve fixed robotic arms operating under different environmental conditions, while the last features a mobile robot within a closed environment. Detailed examples of robotics video ysamples can be found in Figures B.10 to B.14. We represent the temporal causal structures of these five scenes using encapsulated temporal causal graphs. A more intuitive and detailed overview, including data size, causal graph, and description, is provided in Table B.4.

## B.6    Traffic situation analysis

For the Traffic Situation Analysis domain, we provide five distinct scenes corresponding to traffic conditions across five different urban maps. Here, we use the indicators Town01, Town02, Town04, Town05, and Town10 to represent different city environments. The detailed examples of traffic videos in these maps can be found in Figure B.15 to Figure B.19. These five scenes share a similar driving logic for vehicles, which allows them to utilize a common causal structural graph. However, the distribution of environmental noise differs significantly in these scenes. Specifically, each city exhibits unique road layouts, varying traffic flow patterns, and differences in both vehicle numbers and pedestrian densities. A more intuitive and detailed overview is provided in Table B.5.

## B.7    Comparative analysis of dataset realism

Bridging the domain gap is crucial for enabling effective real-world generalization from synthetic data. Thus, we qualitatively compare our dataset with standard synthetic alternatives. Compared with standard synthetic datasets, our collection narrows the gap between synthetic and real-world data by explicitly increasing both rendering granularity and scene realism.

**Image data comparison**    We take the domain of static image generation as an example to be compared with other synthetic datasets, such as Causal3DIdent [16]. Figure B.20 presents a side-by-side comparison between our static images and those in Causal3DIdent [16]. In contrast to the simple backgrounds and single-light sources that dominate most synthetic datasets, we introduce complex scenes together with multiple light types during rendering, which reproduces more realistic illumination and richer background detail. In addition, we rely on high-resolution assets and render every image in $1024 \times 1024$ pixels, resulting in a sharper appearance and finer textures.

**Video data comparison**    Taking the domain of traffic situation analysis as an example, the use of Unreal Engine 4 combined with a carefully curated city asset library allows realistic simulation of urban traffic. The native weather system of Unreal Engine 4 further permits diverse and dynamic atmospheric conditions, increasing environmental realism beyond that of existing synthetic video datasets. Figure B.21 contrasts frames from our traffic scenes with those in Cloth [3]. Our data demonstrates greater visual complexity and realism, offering a more faithful approximation of real-world applications.

Table B.3: Data size, causal graph, and description for dynamic physical process analysis.

| Scene | Data Size | Causal Graph | Description |
|-------|-----------|--------------|-------------|
| Fall

Figure B.7 | ~29k |  | This scene simulates the free fall motion of an object. The temporal transition function defines the state evolution over time as follows: $v_{t+1,1} = v_{t,1} + g \cdot \Delta t,$ $h_{t+1,1} = h_{t,1} + v_{t,1} \cdot \Delta t + 0.5 \cdot g \cdot (\Delta t)^2,$ where $v_{t,1}$ denotes the velocity in the first dimension (i.e., the vertical $y$-axis) at time step $t$, under a 3D Cartesian coordinate system. $h_{t,1}$ represents the height of the object at time $t$, which is associated with the vertical coordinate $x_{t,1}$. For convenience, we will use coordinates for mathematical formulation hereafter. As the partial causal graph of **Fall Simple**, this serves as a simplified version of **Fall Complex** to some extent. |
| Projectile

Figure B.8 | ~29k |  | This scene simulates the horizontal projectile motion of an object. The temporal transition function defines the state evolution over time as follows: $v_{t+1,1} = v_{t,1} + g \cdot \Delta t,$ $x_{t+1,1} = x_{t,1} + v_{t,1} \cdot \Delta t + 0.5 \cdot g \cdot (\Delta t)^2,$ $v_{t+1,0} = v_{t,0},$ $x_{t+1,0} = x_{t,0} + v_{t,0}.$ |

Table 4: Data size, causal graph, and description for dynamic physical process analysis.

| Scene | Data Size | Causal Graph | Description |
|-------|-----------|--------------|-------------|
| Collision

Figure B.9 | ~30k | 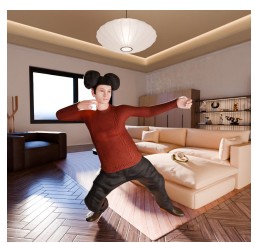 | This scene simulates the collision motion of two objects. The temporal transition function defines the state evolution over time as follows: $v_{t+1,0}^{(1)} = \frac{(m^{(1)} - m^{(2)})v_{t,0}^{(1)} + 2m^{(2)}v_{t,0}^{(2)}}{m^{(1)} + m^{(2)}}$, $v_{t+1,0}^{(2)} = \frac{(m^{(2)} - m^{(1)})v_{t,0}^{(1)} + 2m^{(1)}v_{t,0}^{(1)}}{m^{(1)} + m^{(2)}}$, $x_{1,0}^{(t+1)} = x_{1,0}^{(t)} + v_{t,0}^{(1)} \cdot \Delta t$, $x_{2,0}^{(t+1)} = x_{2,0}^{(t)} + v_{t,0}^{(2)} \cdot \Delta t$. |

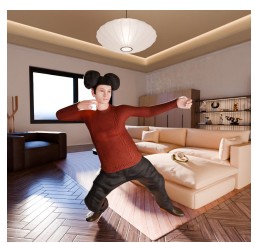
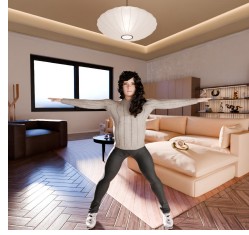
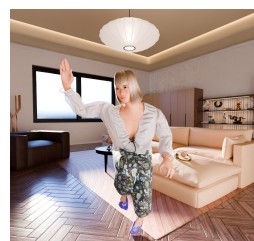
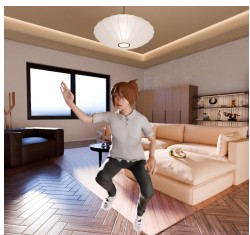
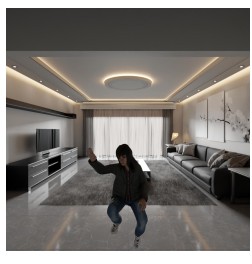
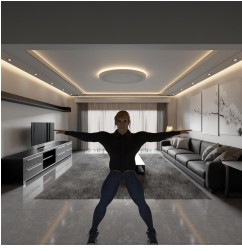
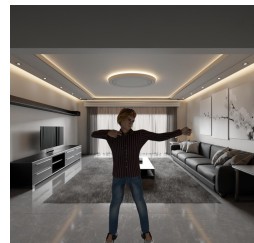
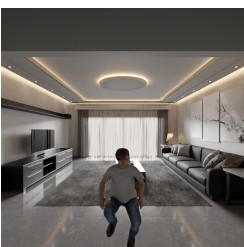

Figure B.1: **Example images from the two indoor scene categories: Human in Living Room and Human in Modern Apartment.** The variations across scenes affect only the environmental context, like background and light conditions.

Table B.4: Data size, causal graph, and description for robotics analysis.

| Showcase | Data Size | Causal Graph | Description |
|---|---|---|---|
| Figure B.10 to Figure B.14 | ~18k | 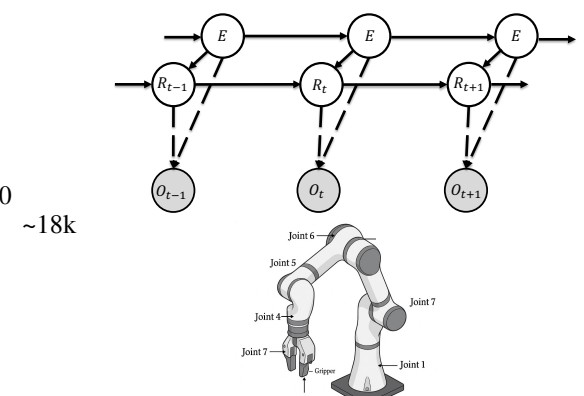 | The figure abstractly illustrates the interaction processes between the robot, the environment, and objects across five different robotics scenes. Here, $E$ encapsulates global environmental factors such as table texture, material, and lighting variations. $R$ abstractly represents the robot's internal states induced by its actions during interactions. These states include joint positions reflecting the movement of multiple robot arm joints, gripper position, end-effector position, and the robot's orientation (rotation), among others. In the Robotics Mobile scene, $R$ also involves significant changes in the robot's overall position, whereas in the other scenarios, the robot's position remains fixed. As shown in the figure, in fixed settings, the robot's joints and end-effector (ee) exhibit diverse motion patterns over time, resulting in rich video dynamics. Finally, $O$ encapsulates the positions and rotations of manipulable objects in the environment, representing the interaction between the robot and these objects. |

Table B.5: Data size, causal graph, and description for traffic situation analysis.

| Showcase | Data Size | Causal Graph | Description |
|---|---|---|---|
| Figure B.15 to Figure B.19 | ~33k | 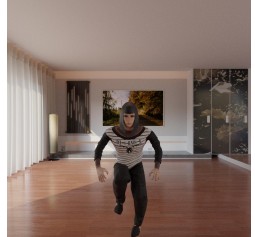 | The left panel depicts a **partial causal graph** obtained after abstracting the latent variables. All traffic lights in the scene are collectively represented as $Lit_i$, the operational states of every vehicle (including throttle, brake, and steering .etc) are represented as $Ani_i$, the orientation of each vehicle is represented as $Rot_i$, and the position of each vehicle is represented as $Loc_i$, where $i$ denotes the current time step. Please note that the variables and relations are more complex in the real case than in the left partial graph. |

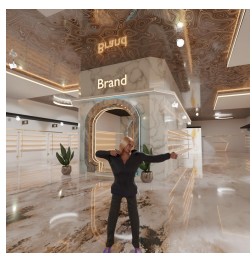 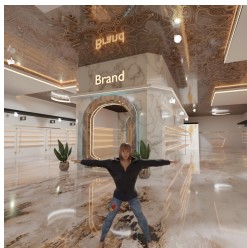 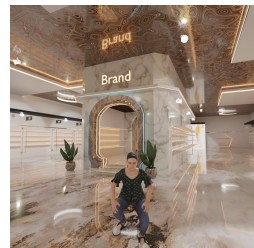 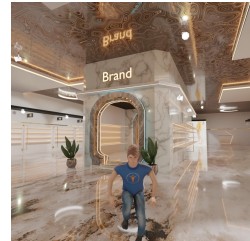

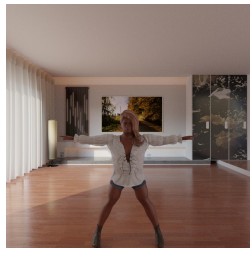 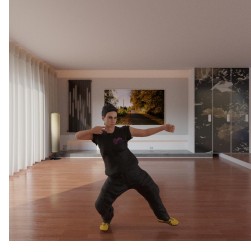 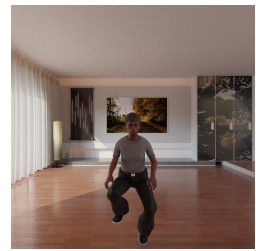 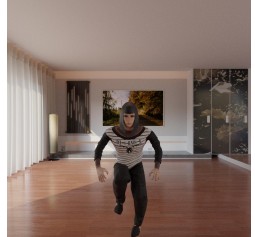

Figure B.2: **Example images from the two indoor scene categories Human in Retail Store and Human against Background Wall.**

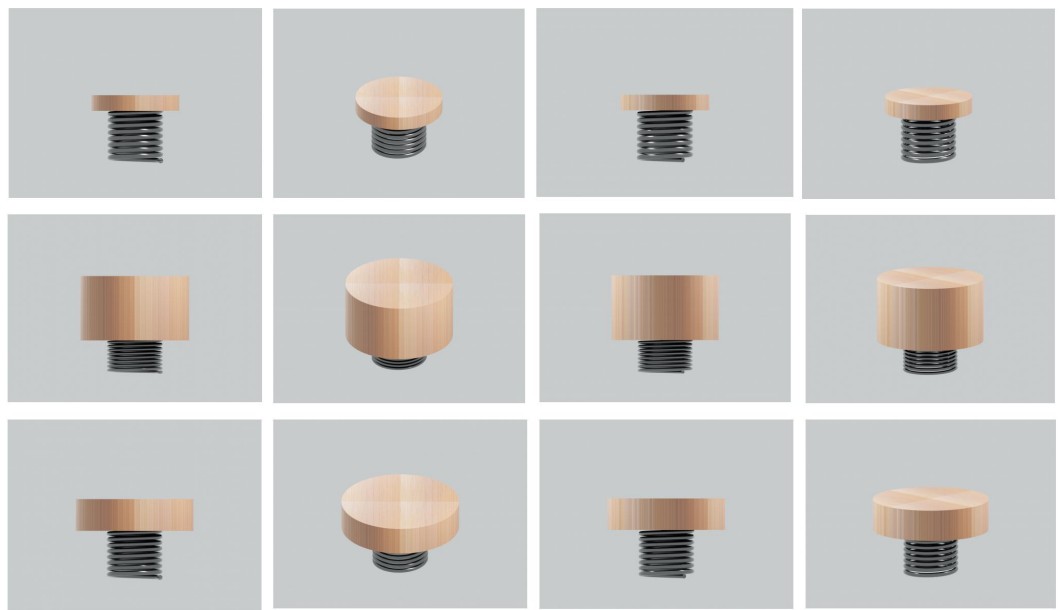

Figure B.3: **Sample frame from the Cylinder Spring scene.** It shows a homogeneous cylinder compressing an ideal spring under gravity until static equilibrium is reached.

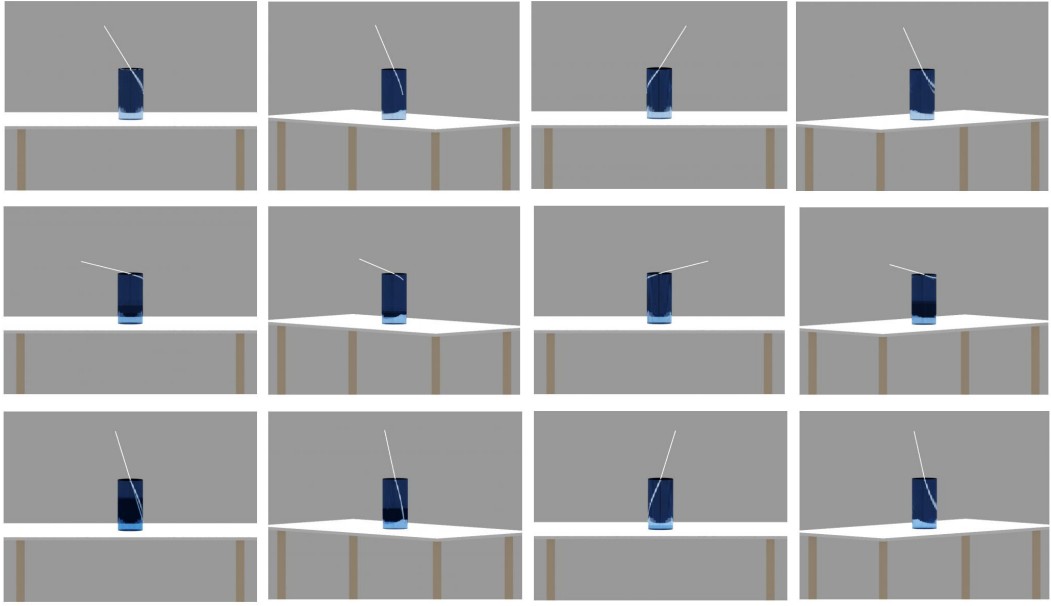

Figure B.4: **Sample frame from the Light Refraction.** This scene depicts a collimated light ray refracting at the interface between air and an aqueous ink solution of varied refractive index, following Snell's law.

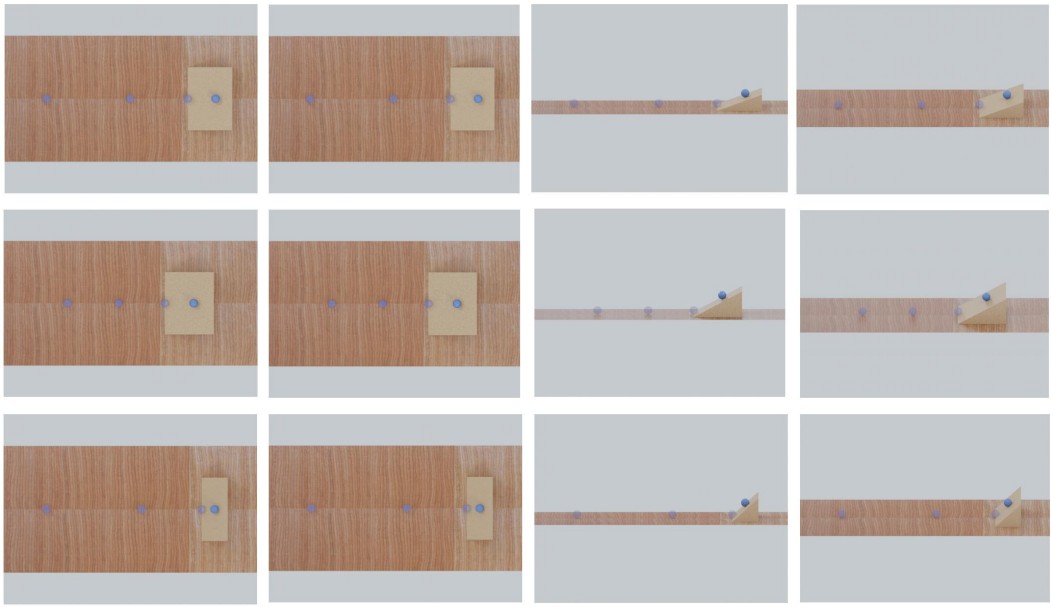

Figure B.5: **Sample frame from the Ball on the Slope scene.** It illustrates a sphere decelerating across a flat surface due to friction and then ascending an incline, with travel distance governed by initial speed, incline angle, and surface roughness.

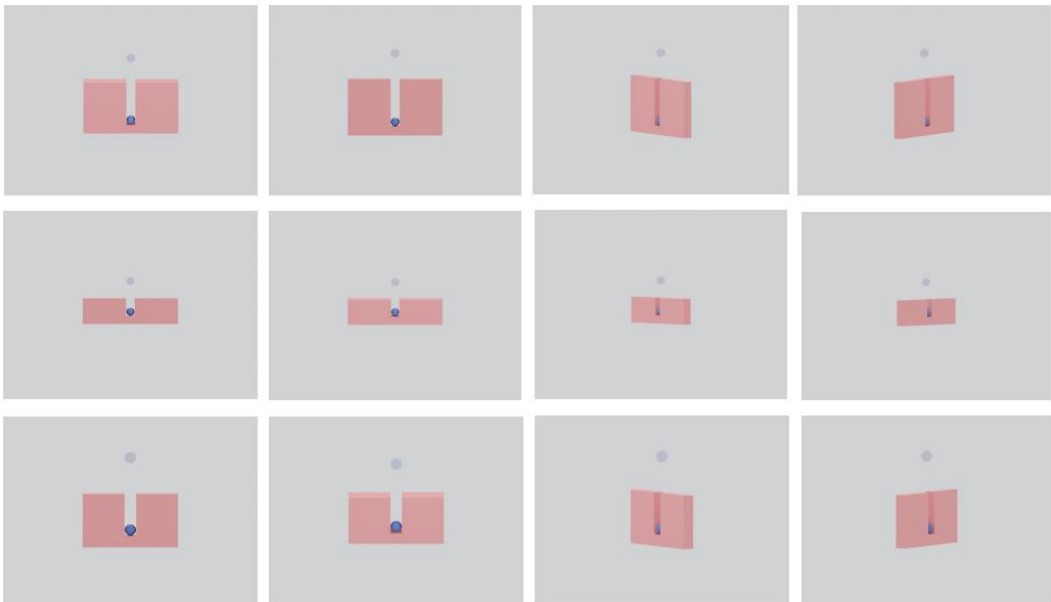

Figure B.6: **Sample frame from the Ball Meets Plasticine scene.** This sense shows a sphere in free fall impacting a viscous clay medium, with penetration depth determined by its mass and the medium's viscosity.

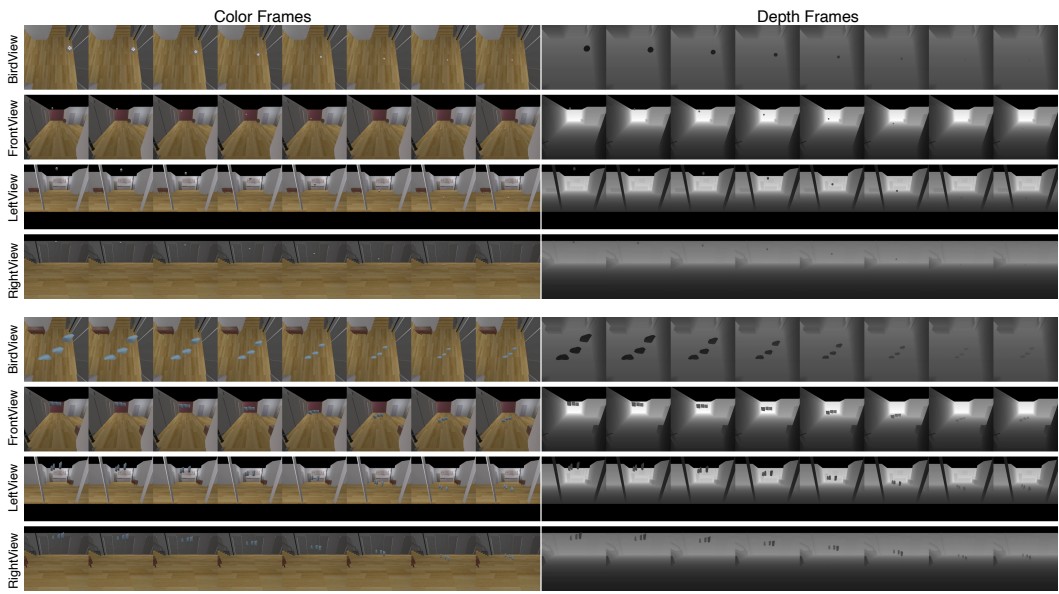

Figure B.7: **Visualization of two samples from the Fall Complex scene.** The two samples depict free-fall scenarios involving a single object and multiple objects, respectively. For each sample, four viewpoints—birdview, frontview, leftview, and rightview—are arranged in rows. Within each row, we present color frames and depth frames captured by different sensors from the same viewpoint. Each sensor-view pair includes 4 frames, sampled at a consistent rate within each sample, with one-to-one correspondence between color and depth frames.

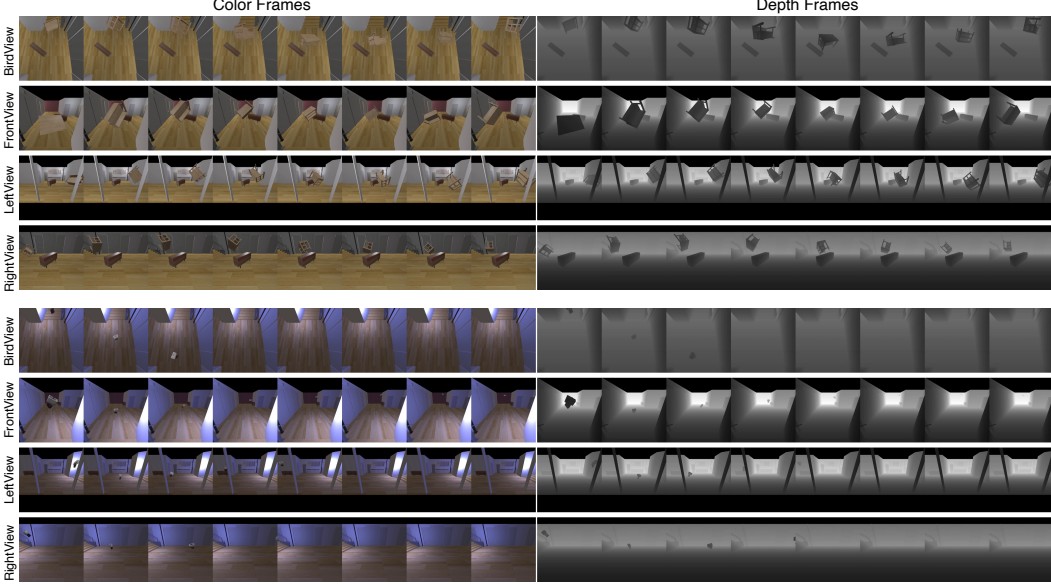

Figure B.8: **Visualization of two samples from the Projectile Complex scene.** Sample 1 illustrates a complex projectile motion of a large object (a table) exhibiting rotational dynamics. Sample 2 showcases projectile motion under varying scenes and lighting conditions. For each sample, four viewpoints—birdview, frontview, leftview, and rightview—are arranged in rows. Each row presents color frames and depth frames captured by different sensors from the same viewpoint. Each sensor-view pair includes 4 frames, sampled at a consistent rate within each sample, with one-to-one correspondence between color and depth frames.

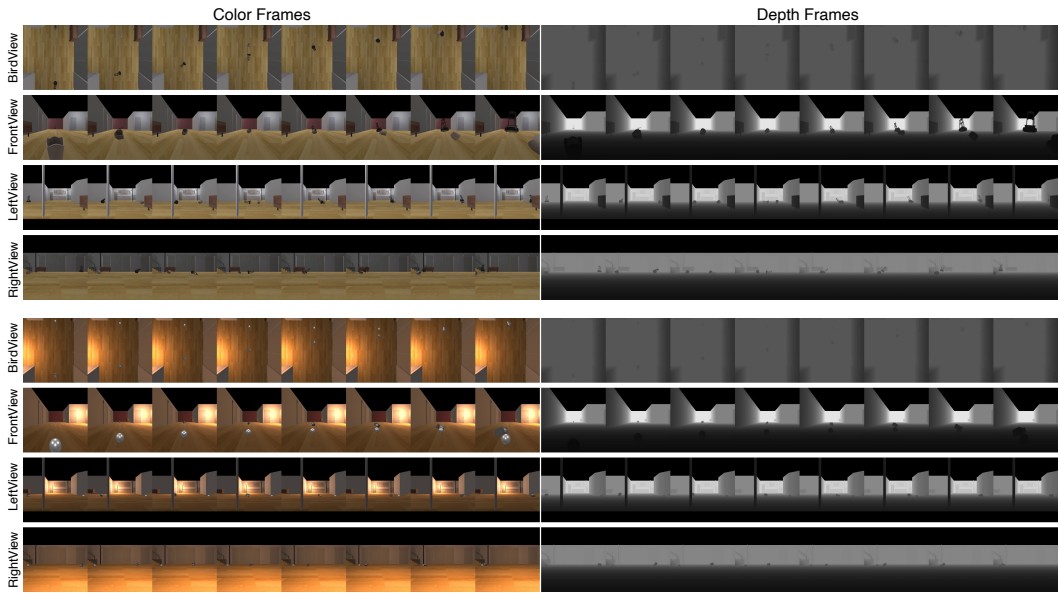

Figure B.9: **Visualization of two samples from the Collision Complex scene.** The two samples depict free-fall scenarios involving a single object and multiple objects, respectively. For each sample, four viewpoints—birdview, frontview, leftview, and rightview—are arranged in rows. Within each row, we present color frames and depth frames captured by different sensors from the same viewpoint. Each sensor-view pair includes 4 frames, sampled at a consistent rate within each sample, with one-to-one correspondence between color and depth frames.

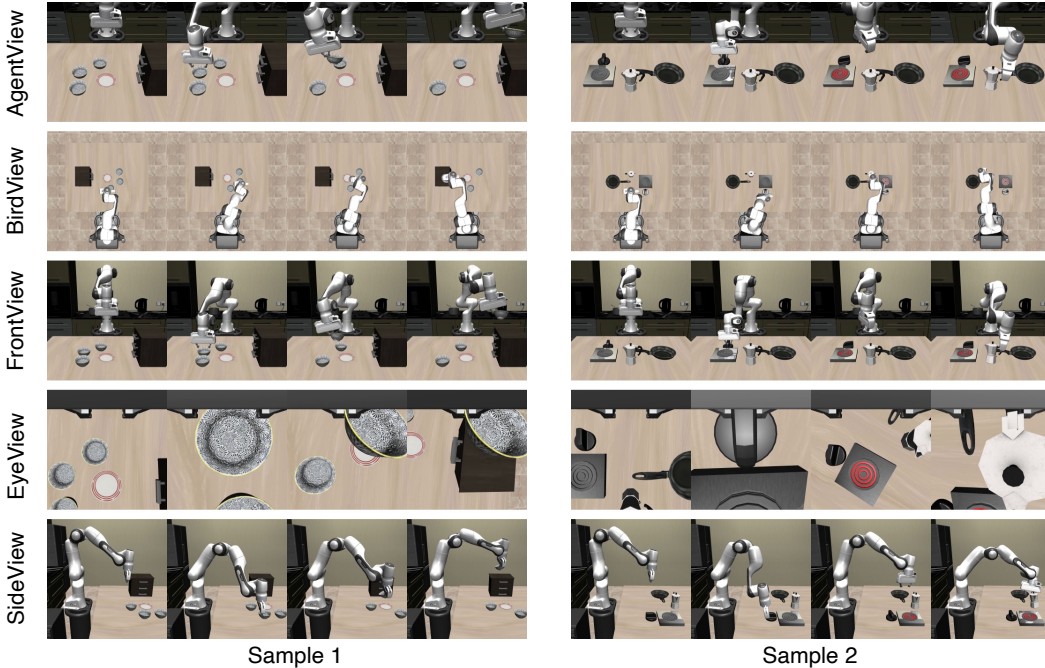

Figure B.10: **Visualization of two samples from the Robotics Kitchen scene.** Sample 1 shows the robot grasping a small bowl on the table, while Sample 2 illustrates the robot attempting to grasp a kettle on the table — potentially for subsequent actions such as reclassification, placing it near a heating area, or even failure. For each sample, five viewpoints are arranged in rows. Each sample is temporally sampled at regular intervals starting from t = 0, and different views at the same timestep provide complementary view-specific information about the robotic arm's actions.

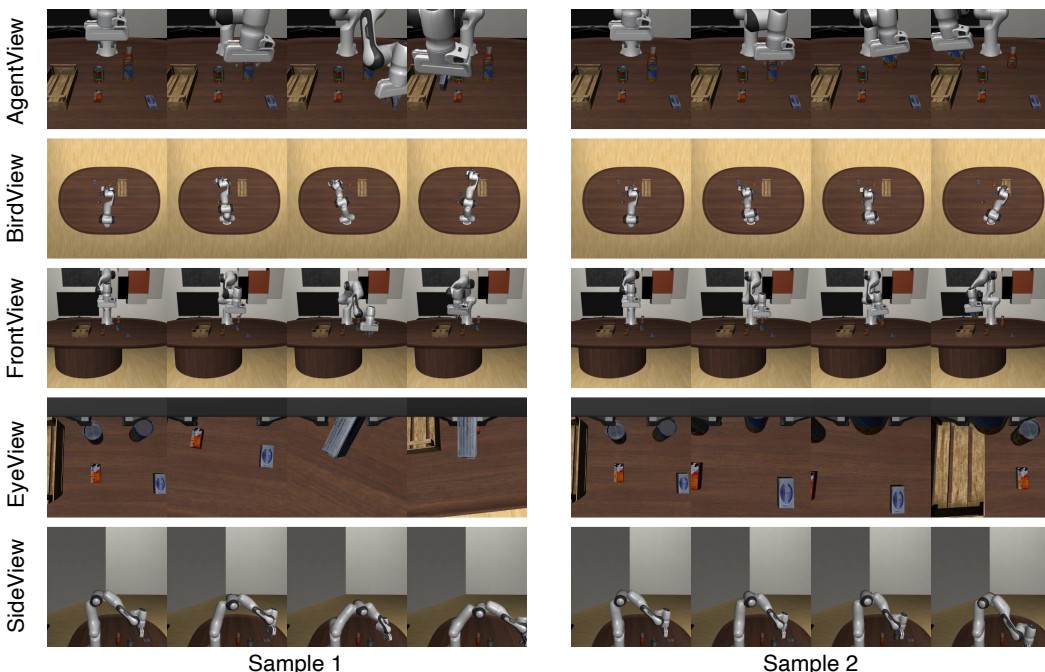

Figure B.11: **Visualization of two samples from the Robotics Living scene.** This scene captures various actions performed by a robot in a home living room environment. Sample 1 shows the robot grasping a book on the table, while Sample 2 depicts the robot attempting to grasp a beverage can on the table, potentially for later organization. For each sample, five viewpoints are arranged in rows. Each sample is temporally sampled at regular intervals starting from t = 0, and different views at the same timestep provide complementary view-specific information about the robotic arm's actions.

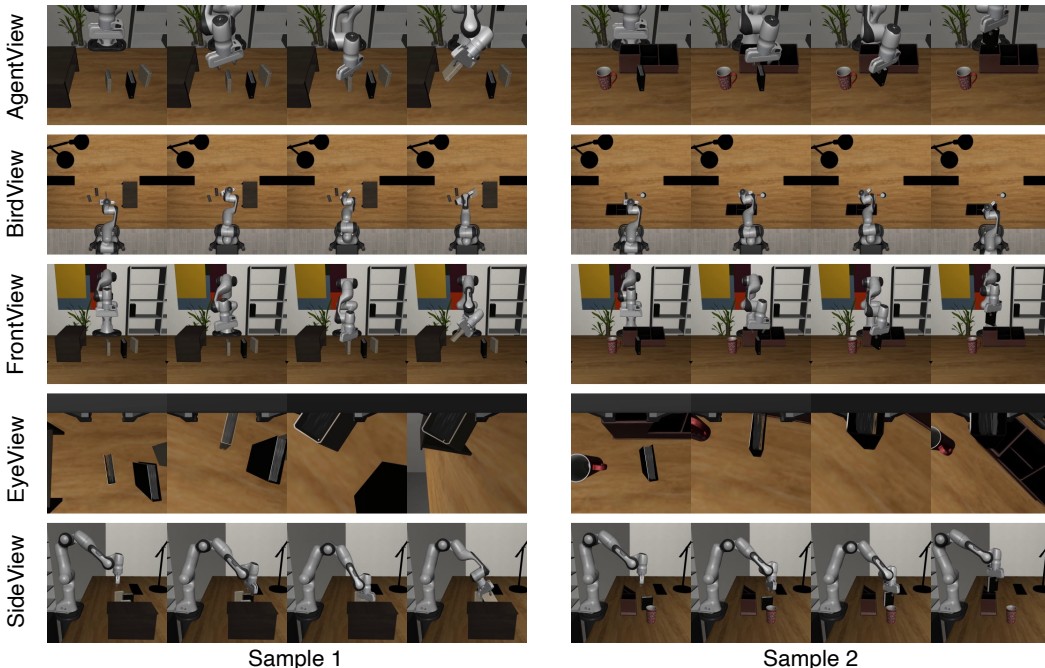

Figure B.12: **Visualization of two samples from the Robotics Study scene.** This scene showcases various actions performed by a robot in a study room environment, where a wide variety of objects are placed on the table. Sample 1 illustrates the robot grasping a book from the table and attempting to place it on a small bookshelf. Sample 2 shows the robot trying to grasp another book from the table, potentially for later organization. For each sample, five viewpoints are arranged in rows. Each sample is temporally sampled at regular intervals starting from t = 0, and different views at the same timestep provide complementary view-specific information about the robotic arm's actions.

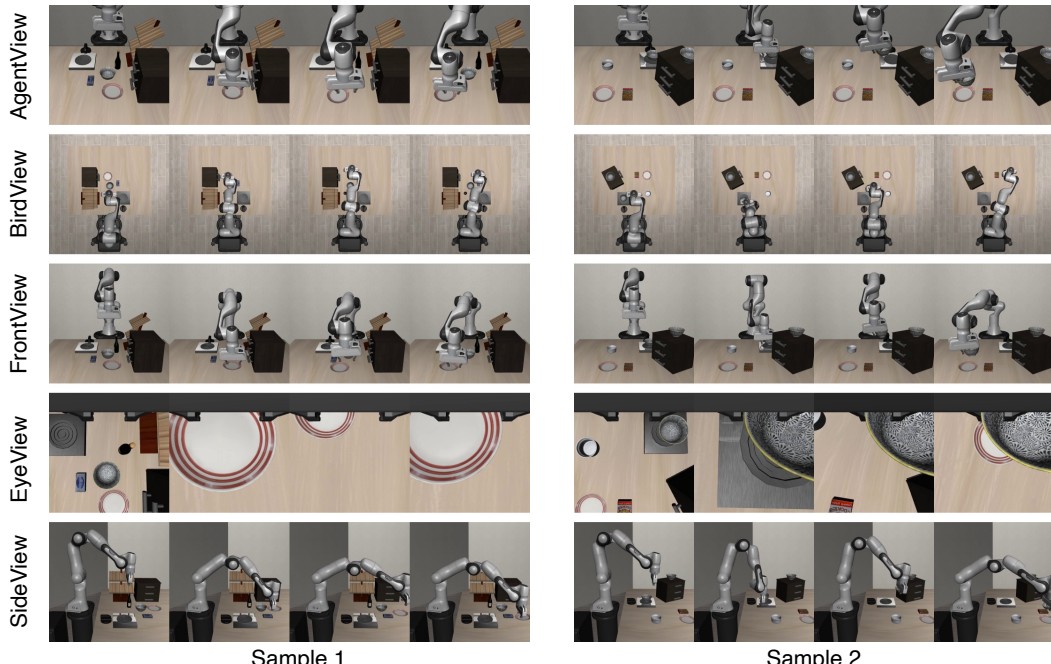

Figure B.13: **Visualization of two samples from the Robotics General scene.** This scenario captures various actions of a robot operating in an open and general environment (i.e., not limited to a specific household setting, hence the name General). Sample 1 shows the robot grasping a small bowl on the table, while Sample 2 illustrates the robot attempting to grasp a container on the table. For each sample, five viewpoints are arranged in rows. Each sample is temporally sampled at regular intervals starting from t = 0, and different views at the same timestep provide complementary view-specific information about the robotic arm's actions.

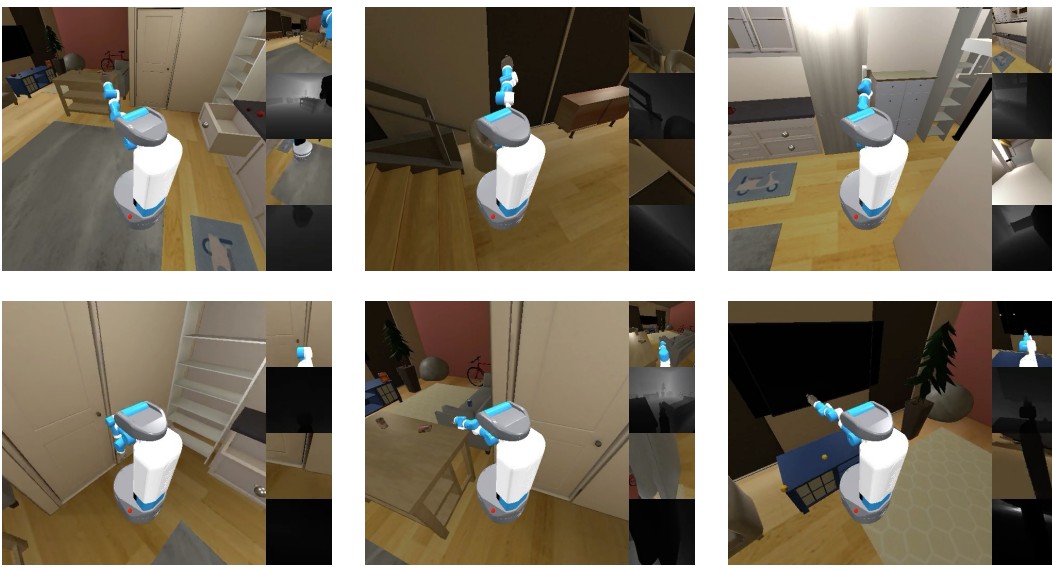

Figure B.14: **Visualization of six frames from the Robotics Mobile scene sampled from one sequence to illustrate temporal continuity.** The scene depicts a robot moving in all directions and waving its robotic arm in a confined environment. For each frame, we concatenate a 512×512 third-person video frame of the main agent with four 128×128 frames captured from two additional agent perspectives (including both color and depth views), forming a complete frame. Users can selectively extract and utilize different parts according to their specific needs.

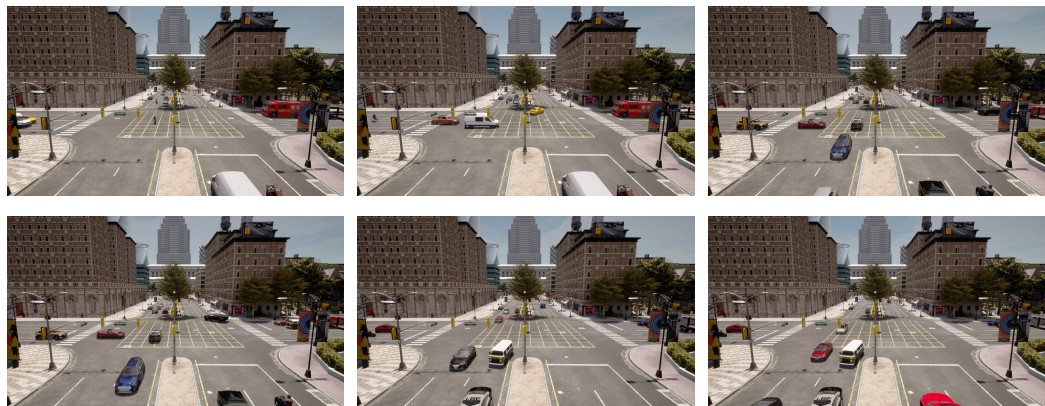

Figure B.15: **Example traffic scene video captured in Town10.** Six frames are sampled from one sequence to illustrate temporal continuity. Although global variables such as traffic density differ among cities, the fundamental driving logic of vehicles remains unchanged.

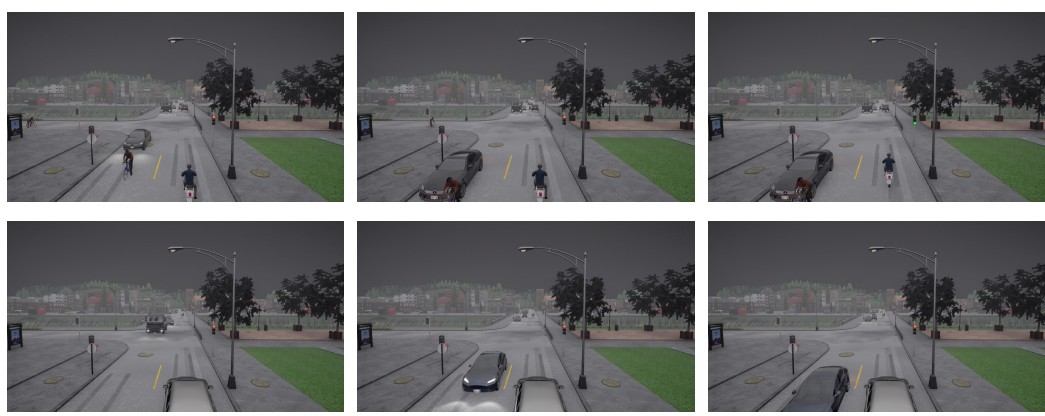

Figure B.16: **Example traffic scene video captured in Town01.**

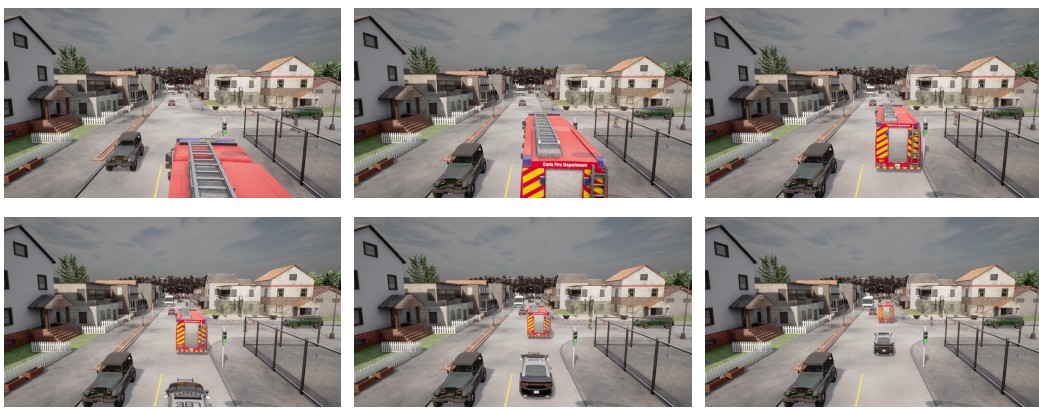

Figure B.17: **Example traffic scene video captured in Town02.**

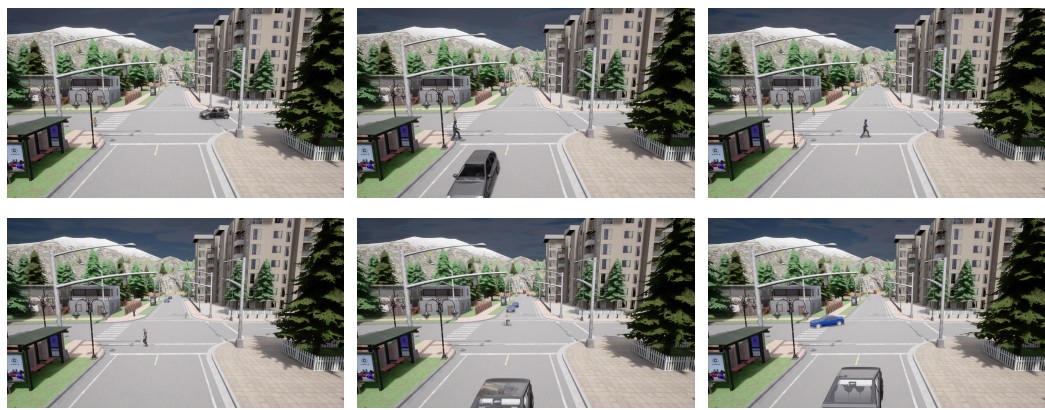

Figure B.18: **Example traffic scene video captured in Town04.**

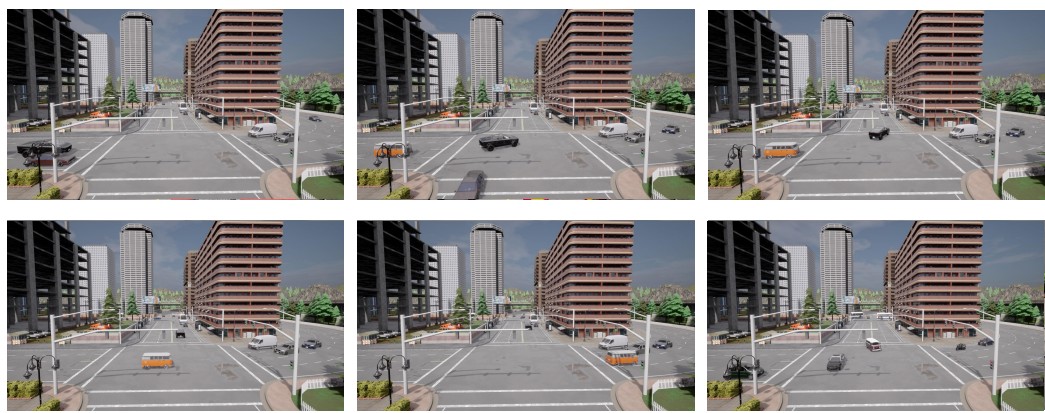

Figure B.19: **Example traffic scene video captured in Town05.**

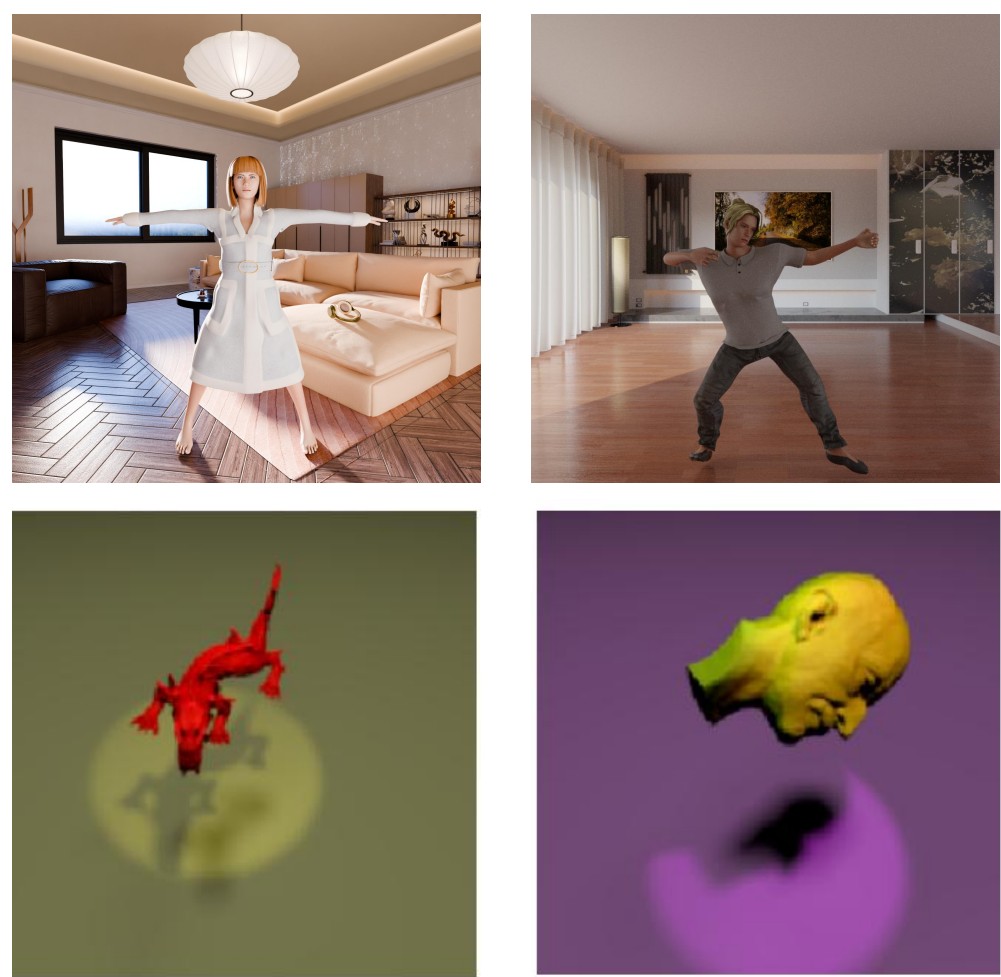

Figure B.20: **Comparison of image realism.** The top row shows static images from our dataset rendered in complex scenes with multiple light sources, while the bottom row shows images from the Causal3DIdent dataset rendered against a uniform background with a single light source.

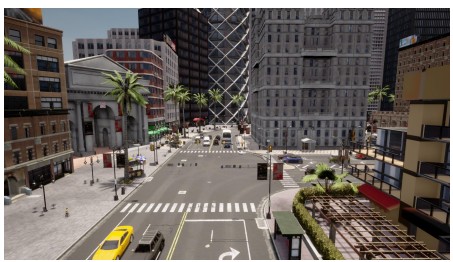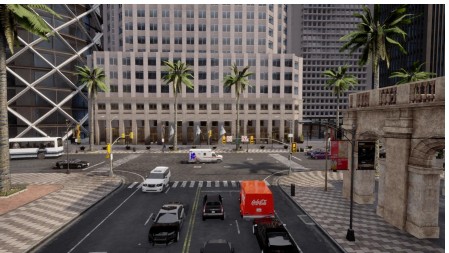

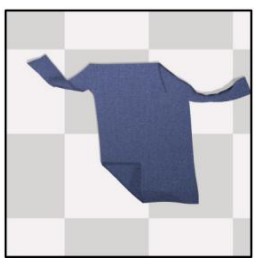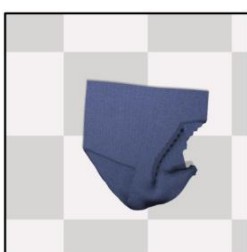

Figure B.21: **Comparison of video data realism.** The top row shows two video frames from the Traffic Situation domain of our dataset, featuring realistic and complex urban scenes populated by numerous interacting agents. The bottom row presents two frames from the Cloth [3] dataset, which uses synthetic backgrounds and depicts only a single simple object per frame.

## B.8 An scene example detailing causal variables and relations

We walk through one concrete dataset example to provide visual and conceptual intuition about the elements we expose—namely, the causal variables and their relationships. We illustrate this using the *Fall Simple* scene. In this scene, global variables specify time–invariant properties of the setup, while dynamic variables evolve across simulation steps, and observation variables record what a sensor would see. The global layer includes the scene identifier (which selects assets and camera placement), the object's category (e.g., cube, sphere), and the acceleration of gravity. The dynamic layer contains the object's 3D position and rotation at each exported time step $t \in \{0, \ldots, T - 1\}$. Under a constant gravitational field, the vertical component of velocity increases approximately linearly with time—formally, $v^{(t+1)} = v^{(t)} + g \Delta t$ and $y^{(t+1)} = y^{(t)} + v^{(t)} \Delta t + \frac{1}{2} g (\Delta t)^2$—so the vertical displacement between successive frames grows as the object accelerates downward. The observation layer then renders these states to RGB (and, where applicable, depth) frames using the selected scene context. Causally, gravity influences vertical acceleration, which in turn updates velocity and position over time; initial conditions propagate forward through these update equations; the object category selects a rendering asset that affects appearance but does not modify the physical law; and the scene context (background geometry and lighting) affects only the rendering of pixels and not the underlying state transitions. This separation lets us vary context and appearance without confounding the physical mechanism, while still producing rich visual diversity.

| Category | Sub-category | Variable | Dim. | Type | Range | Description |
|---|---|---|---|---|---|---|
| Global | Scene | scene | (1,) | D | 6 types | Scene identifier (assets, camera layout) |
| Global | Scene | gravity | (1,) | C | – | Acceleration of gravity $(\mathrm{m\,s^{-2}})$ |
| Global | Object | render_asset | (1,) | D | 90 types | Visual appearance of the falling object |
| Dynamic | Object | position | (T,3) | C | – | 3D coordinates across exported time steps |
| Dynamic | Object | rotation | (T,3) | C | – | Euler angles across exported time steps |
| Dynamic (derived) | Object | velocity | (T,3) | C | – | Finite-difference estimate using $\Delta t$ |
| Dynamic (derived) | Object | accel_y | (T,1) | C | – | Vertical acceleration; equals $g$ under no drag |

**Notes.** $T$ denotes the number of exported frames; *Type:* D = discrete, C = continuous. Derived variables are computed from the primary dynamic states and the export step $\Delta t$; they may be provided directly or recomputed from the released states.

Details regarding the data size, causal graph, and description for dynamic physical process analysis can be found in Table B.3. We also include a video showcase for this and other scenes in the documentation and on our webpage.

## B.9 Flexible configurations

This subsection details how the dataset exposes flexible configurations of the underlying causal generative process while preserving a clear separation between structure, parameters, and observation protocol. The controls cover visually grounded domain labels, temporal dependencies that govern how present states evolve from the past, and explicit intervention histories that can be logged and replayed. Together, these levers allow researchers to create targeted conditions that either satisfy common assumptions in causal representation learning or deliberately violate them in a controlled manner.

First, in the static image domain, changing the domain identifier switches the background geometry and high dynamic range lighting without altering the semantic latent factors that generate the human subject. Practically, the same identity, pose, clothing, and body attributes are rendered in visually distinct environments with different illumination patterns, color temperatures, occlusions, and clutter statistics. This isolates contextual variation as a pure domain shift while the latent causal graph for the

subject remains unchanged. Such a setting is particularly useful when evaluating sensitivity to context or when treating scenes as separate domains in adaptation protocols, because the supervision available to the learner is the same set of factors even though the pixel distribution differs substantially.

Second, in the free–fall video scene, varying the gravitational constant across a small, interpretable grid, for example $g \in \{4.9, 9.8, 14.7\}\,\mathrm{m\,s^{-2}}$, yields domains that differ only in the strength of the mapping from height to velocity. The transition is governed by standard kinematics with time step $\Delta t$, namely $v^{(t+1)} = v^{(t)} + g\,\Delta t$ and $y^{(t+1)} = y^{(t)} + v^{(t)}\Delta t + \frac{1}{2}g(\Delta t)^2$, while the horizontal velocity remains constant in the absence of drag. Changing $g$ therefore adjusts the temporal rate at which potential energy converts to kinetic energy without modifying the graph structure or introducing additional confounders. Because the modification is parametric and physically interpretable, it supports crisp counterfactual statements such as "under the same initial conditions, the object would reach the ground earlier or later solely due to a different gravitational field."

Third, in the traffic videos, behavioral rules translate directly into temporal dependencies by altering the stochastic mapping from the previous traffic state to the current action profile. A canonical example is the minimum headway rule (the desired spacing between successive vehicles). Tightening this rule increases the likelihood of braking and lane–keeping behaviors when relative distance shrinks, which can be described as a shift in the conditional distribution $P(A^{(t)} \mid S^{(t-1)})$. Crucially, the high–level causal structure that links environment, agent states, and actions is retained, but the policy governing interactions changes in a measurable, semantically meaningful way. By sweeping the headway target across several levels, one can move smoothly from dense, stop–and–go traffic to free–flow conditions and examine whether a learned representation tracks the underlying decision mechanisms rather than surface statistics.

Fourth, temporal dependencies can also be manipulated through the observation protocol by adjusting the recording time step while keeping the simulator's internal dynamics fixed. Increasing the export cadence from $0.02\,\mathrm{s}$ to $0.10\,\mathrm{s}$ reduces temporal resolution and makes the visible sequence less Markova at the frame level: information that was previously captured by immediate neighbors is now spread across longer lags, so the effective transition kernel in the observed space becomes higher order even though the latent physical process is unchanged. This configuration probes whether temporal representation learners identify causal factors that are stable to sampling changes, a property that matters in practice whenever sensors operate at heterogeneous frame rates or logs are down-sampled for storage.

Fifth, intervention histories are supported both in robotics and in physics scenes, enabling precise do–style manipulations and deterministic replay. In robotics, a scripted, deterministic motion such as a full–arc sweep imposes a hard intervention on the action channel; each call is timestamped so that the same initial state and random seed reproduce the identical trajectory for counterfactual comparison against an alternative script. In physical environments, mechanism parameters like the gravitational constant, the spring stiffness, or the restitution coefficient can be overwritten at designated frames to realism either soft interventions that momentarily perturb the mechanism or hard interventions that reset and hold it thereafter. Because these edits are localized in time and recorded alongside the sequence, one can align pre– and post–intervention segments, quantify the causal effect on latent variables and observables, and evaluate whether the learned representation follows the intervened mechanism rather than spurious correlations.

Each control is designed to be minimal with respect to the causal graph (structure held fixed whenever possible) while still inducing a detectable and interpretable change in the data–generating process, thereby supporting fine–grained diagnosis of model behavior. We just show some examples of flexible configurations and research can use the ground truth of causal graphs to do further configurations. Besides, we have released all the source codes and documents of data generation so researchers can create their own data to further corroborate their experiments.

## C  Experiments

### C.1  Image-based method implementation details

**Model design**     To fairly compare the performance of different CRL methods on our dataset, we select four representative unsupervised approaches and one supervised upper bound: **Sufficient Change** [6] (domain-shift based identifiability), **Sparsity** [41], **Multiview** [16], **Contrastive Learning** [42], and

**Supervised**. All methods share a ResNet-18 backbone pretrained on ImageNet: we drop its final fully connected layer, apply adaptive average pooling followed by flattening to produce a 512-dimensional feature vector, and then attach a lightweight head. **Sparsity** adopts a variational autoencoder design: the pooled feature passes through two linear–ReLU stages and then splits into parallel linear heads that output the mean $\mu$ and log-variance $\log \sigma^2$ of a latent vector $z$; during training we sample $z \sim \mathcal{N}(\mu, \sigma^2)$ via the reparameterization trick and reconstruct back to 512 dimensions with a symmetric decoder, optimizing an $\ell_2$ reconstruction loss plus $\mathrm{KL}\big[\mathcal{N}(\mu, \sigma^2) \,\|\, \mathcal{N}(0, I)\big]$. **Multiview** attaches a three-stage MLP projection head (linear–ReLU–linear–ReLU–linear) that maps the 512-dimensional feature into a compact embedding whose dimensionality is set by the dataset metadata; it is trained with a contrastive objective that pulls together samples sharing the same causal attribute and pushes apart the others. **Sufficient Change** instantiates the sufficient-changes principle [6]. We split the latent space of dimension $d$ into a content subspace $z^{\mathrm{cont}} \in \mathbb{R}^c$ and a style subspace $z^{\mathrm{sty}} \in \mathbb{R}^s$. The style branch is passed through a conditional normalizing flow whose parameters are generated by an auxiliary MLP from a learned domain embedding $u$. The flow output is concatenated with $z^{\mathrm{cont}}$ to form a deconfounded latent $\tilde{z}$, which is used both for reconstruction via a symmetric decoder and for classification via a downstream MLP head. The total loss jointly optimizes the evidence lower bound, the flow log-determinant, and a classification cross-entropy term. **Contrastive Learning** follows a SimCLR-style instance discrimination realization [42]: a projection MLP (linear–ReLU–linear) maps the 512-dimensional feature to an embedding of dimension $d^{\mathrm{proj}}$ used by an InfoNCE loss with two stochastic augmentations per image. Positives are two views of the same image; all other in-batch views serve as negatives. At evaluation, the projection head is discarded and the pre-projection representation is used as the learned latent. **Supervised** is a purely supervised variant: after pooling to 512 dimensions it applies a simple MLP (linear–ReLU–linear–ReLU–linear) to produce a $d$-dimensional embedding trained with a supervised regression objective, without any reconstruction or latent regularization.

**Loss and data arrangement**   All models train on the same metadata-annotated image collection, which is split once into training and test subsets with a fixed ratio. Mini-batch formation differs by method: **Sparsity** and **Supervised** use standard random sampling of individual images; **Multiview** samples images at random and applies two stochastic augmentations per image to form positives according to the shared causal attribute; **Contrastive Learning** also uses two augmentations per image but defines positives as two views of the same image and uses all other in-batch views as negatives; **Sufficient Change** replaces random batching with a `BalancedBatchSampler` so that each batch contains an equal number of examples from each of the four view domains. In optimization, **Sparsity** minimizes a VAE reconstruction plus KL divergence loss (weighted by $\lambda^{\mathrm{VAE}}$), a Jacobian sparsity penalty (weighted by $\lambda^{\mathrm{sparsity}}$), and a mean squared error on the latent codes; **Multiview** optimizes a contrastive objective over pairwise augmented views defined by shared causal attributes; **Contrastive Learning** minimizes an InfoNCE loss with temperature $\tau$ over instance-discrimination pairs; **Sufficient Change** jointly minimizes a per-domain VAE loss (reconstruction plus a clamped KL term) and a Gaussian-prior log-likelihood penalty on the flow-transformed style subspace (weighted by $\lambda^{\mathrm{gauss}}$); and **Supervised** uses a regression loss on its final embeddings. All methods employ identical Adam settings (same learning-rate schedule and weight decay) and report per-epoch global Pearson MCC and uniform-average $R^2$ on both training and test splits.

### C.2   Video based method implementation details

**Model design**   Due to the high resolution of videos in our datasets—even the smallest scenes reach $512 \times 512$—we employ pretrained high-quality VAE encoders to convert videos into compact superpixel-level representations. This design enables stable baseline training while allowing subsequent temporal causal representation learning (CRL) methods to operate directly on the superpixels with minimal influence.

We explore two pretrained VAEs: a frame-based VAE from Stable Diffusion [65] and a video-based VAE from CogVideoX [66]. Their downsampling factors are $(8, 8, 1)$ and $(8, 8, 4)$ along the $(H, W, T)$ dimensions, respectively. The Stable Diffusion VAE provides temporally independent latents that better preserve per-frame spatial fidelity, while the CogVideoX VAE exhibits stronger long-term temporal reconstruction capability. Under our experimental setup—randomly sampling 16 consecutive frames per video—the Stable Diffusion VAE produces more stable and length-consistent latent representations, making it our default choice for generating superpixels.

For the inner VAE architecture, we adopt a lightweight convolutional VAE operating on per-frame superpixels. Each superpixel clip is encoded independently and reconstructed frame by frame. The encoder consists of two stride-2 convolutions with bias compensation, generating spatial feature maps for both the mean and log-variance branches. Latent variables are obtained via a standard reparameterization step, and the decoder mirrors the encoder to reconstruct each frame from an explicit spatial latent. We further introduce Bias-Compensated Convolution, where a learned per-channel bias term is adaptively adjusted whenever pre-bias activations exhibit excessive negativity. All nonlinearities employ Adaptive LeakyReLU, whose slopes can be dynamically tuned based on layer-wise activation statistics collected by an ActivationAnalyzer. This analysis–adjustment process is optional and does not alter the forward computation during training. These two mechanisms improve numerical stability without changing the overall architecture. The model processes frames independently over time; temporal dynamics and causal dependencies are entirely modeled by CRL methods built atop this baseline representation.

The rationale for using this inner VAE configuration is to maximize the recovery of latent variables. In searching for the optimal model, we aim to achieve observational equivalence—that is, to make the reconstructed video as close as possible to the ground-truth video. When we increase the network depth (e.g., using more than three strided convolutional layers) to achieve stronger compression, the model inevitably loses temporal details and background information, leading to reconstructions that fail to preserve human-perceptible motion dynamics. The situation worsens when employing pooling or high-compression MLP-based designs, where the reconstructed video may retain overall structure but appears static, lacking any meaningful temporal variation. Alternative architectures such as U-Net offer skip connections that pass high-resolution features from the encoder to the decoder, helping preserve fine spatial details and enabling highly accurate reconstructions. However, these skip pathways also leak excessive information to the decoder without proper selection, making the estimated latent variables entangled and semantically ambiguous. Consequently, the encoder's representations lose their exclusive explanatory power—a crucial property for causal modeling.

Therefore, our chosen architecture strikes a balance between preserving dynamic temporal information and maintaining static spatial detail. It serves as an effective and interpretable framework for video-based causal representation learning, providing both reconstruction fidelity and a semantically plausible latent space.

**Loss and data arrangement**    All models are trained on the same metadata-annotated video collection, which is split once into training and test subsets with a fixed ratio. For iVAE, the model performs reconstruction while keeping the latent variables close to the domain-conditioned Gaussian prior $p(z \mid u)$. It penalizes dependencies between latent dimensions using mutual information and total correlation terms, with their weights aligned with those of the KL divergence and reconstruction losses. For TCL, latent features from consecutive time steps are treated as positive pairs, while another frame sampled from a different time step forms a negative pair. Both pairs are concatenated and passed through a lightweight temporal discriminator. The contrastive loss weight is set to 0.1. For TDRL and CaRiNG , we set the latent space is eight-dimensional (z_dim = 8), all of which are treated as time-dependent (z_dim_fix = 8, z_dim_change = 0). The transition prior looks back two steps (lag = 2) and uses temporal embeddings of size 8 for 16 discrete time indices (nclass = 16, embedding_dim = 8) to capture dynamics. The networks inside the encoder and priors have a hidden width of 128 (hidden_dim = 128). The training loss is weighted key hyperparameters: gamma = 0.0075 enforces consistency with the Laplacian transition prior for future states. For IDOL, an additional sparsity weight with 0.2 encourages sparsity in the Jacobian of instantaneous and historical influences, promoting interpretable temporal structure. For other hyperparameters, the pretrained VAE yields 4 channels. The inner VAE uses a convolutional kernel size of 4 and a stride of 2. The learning rate is set to 1e-4 for iVAE and TCL, and 1e-5 for the remaining methods.

# D   Statement

CausalVerse provides a high-fidelity, open-source benchmark for causal representation learning that combines realistic visual complexity with fully known, configurable causal generating processes. By enabling reproducible evaluation across diverse domains—static images, dynamic physics simulations, robotic control, and traffic scenarios—our dataset accelerates methodological progress, fosters transparent comparison, and lowers the barrier to entry for both researchers and newcomers in CRL.

Because CausalVerse is built on controlled, simulated data and is intended solely as a research tool rather than for direct deployment, we do not identify any immediate negative societal impacts. We anticipate that broad adoption of this benchmark will strengthen the reliability and real world applicability of future CRL methods without introducing adverse consequences.

