# OpenReview forum: "CausalVerse: Benchmarking Causal Representation Learning with Configurable High-Fidelity Simulations"
_NeurIPS.cc/2025/Datasets_and_Benchmarks_Track — NeurIPS 2025 Datasets and Benchmarks Track spotlight_

### Official Review · Reviewer_EXTi · 2025-06-07

**Rating:** 5
**Confidence:** 3

**Summary:**

This paper introduces CausalVerse, a new large-scale benchmark for Causal Representation Learning (CRL) designed to bridge the gap between evaluative precision and real-world complexity. It provides a high-fidelity simulated visual dataset with fully accessible ground-truth causal variables and structures, enabling the rigorous and flexible evaluation of CRL methods across diverse and realistic scenarios. The paper also presents an empirical evaluation of several representative CRL methods on CausalVerse, offering insights into their performance and limitations, particularly when assumptions are not fully met.

**Additional Feedback:**

1. I suggest that authors take an example of a dataset and illustrate its different elements, like causal variables and relationships, for visual intuition about the dataset. This can be added to the supplementary material.

**Dataset Code Accessibility:**

Partly

**Ethical Considerations:**

No, there are no or only very minor ethics concerns

**Final Justification:**

The authors have addressed my concerns, and I have already recommended acceptance of the paper. Thanks!

**Limitations Weaknesses:**

1. A clear pipeline, along with code for generating this dataset, should be released so that researchers could build on this dataset.

2. In addition to the characteristic comparison with existing benchmarks, a comparative study with the simulation approach of existing benchmarks should also be discussed.

3. The paper evaluates a few representative CRL methods. It would have been better to have the authors compare more state-of-the-art methods.

4. The dataset is limited to computer vision only.

**Strengths Contributions:**

1. The paper addresses an important challenge of evaluating CRL methods by providing a benchmark that combines visual realism with ground-truth causal knowledge. The paper's motivation is clearly articulated by outlining the limitations of existing evaluation strategies. The structure is logical, and the main contributions are explicitly summarized, making the paper easy to follow.
2. They released an extensive dataset, CausalVerse -- created using Unreal Engine 4 and Blender, with a large number of images and video frames across a wide variety of scenarios, from static to dynamic, and single- to multi-agent systems. This diversity provides a comprehensive testbed for evaluating the generalizability of CRL methods. THis also provides for users to access and modify the underlying causal structures and simulation parameters. This allows for controlled experiments where theoretical assumptions can be strictly satisfied or intentionally violated to test the robustness of different methods.
3. The authors demonstrate the utility of their benchmark by conducting a systematic comparison of existing CRL methods. This provides valuable insights for practitioners and helps to highlight the current challenges in applying CRL to complex, realistic data.
4. The dataset is made publicly available, which is a significant contribution to the research community and facilitates reproducible research.

---

> ### Author Rebuttal · Authors · 2025-07-31
>
> Dear Reviewer EXTi,
>
> We sincerely appreciate your thoughtful review, recognition of our contributions, and constructive feedback. Your comments on dataset clarity, benchmark utility, and evaluation are truly encouraging. In response to your suggestions, we have added documentation for the dataset generation pipeline, discussed simulation design choices relative to existing benchmarks, and added more evaluation baselines. Please see the point-by-point responses below.
>
> > **Q1**: A clear pipeline, along with code for generating this dataset, should be released so that researchers could build on this dataset.
>
> **A1**: Thank you for the helpful suggestion! We have organized and prepared the dataset generation code along with supporting documentation. However, due to the rebuttal policy, we are currently unable to update the official repository (the Hugging Face project page). We promise that all documentation and code will be released as soon as the rebuttal policy concludes. We are also more than happy to support the community in extending and customizing the dataset for future research. Below, we provide a brief introduction to one data generation pipeline and code as an example:
>
> **Image part of dynamic physical simulation domain**
>
> * Environment Installation
>
> We leverage the open-source `BlenderProc` toolkit, as documented in its official guide.
> (`pip install blenderproc`; an editable setup is `cd BlenderProc && pip install -e .`)
>
> * File Structure
>
> The project ships with four scenes, including Cylinder Spring, Light Refraction, Ball on the Slope, and Ball Meets Plasticine, each driven by a dedicated script (`Spring.py`, `Refraction.py`, `Slope.py`, `Plasticine.py`).
>
> * Generation Scripts
>
> Image synthesis requires only `blenderproc run <scene>.py`.
>
> * Results Example
>
> In the Refraction scene, the script traverses 40 ink-concentration levels and 250 random incident angles (5°–85°); for every combination it rebuilds the scene, renders four 800 × 600 orthographic views at 256 spp, and logs λ, incident and refracted angles, refractive index, and image paths to `metadata.csv`, producing the stated 40 000 samples.
>
> > **Q2**: In addition to the characteristic comparison with existing benchmarks, a comparative study with the simulation approach of existing benchmarks should also be discussed.
>
> **A2**: Thank you for the suggestion, which prompted us to further explore the literature on simulation approaches. We have included a more systematic discussion of these methods in the appendix, and copied it below:
>
> Contemporary CRL datasets emerge from **four distinct simulation approaches** that together trace a realism‑versus‑controllability spectrum. **(i)** At the low‑fidelity extreme, *Pendulum* and *Flow* exemplify analytic toy‑physics worlds whose closed‑form ODEs and minimalist ray‑tracing enable millisecond image generation and pixel‑perfect ground truth. **(ii)** Stepping up in complexity, mass–spring and rigid‑body engines such as MuJoCo, PhysX, and Box2D drive datasets like *Ball* and *Cloth*, enriching dynamics with contact, friction, and deformable bodies. **(iii)**  A different flavour appears in task‑oriented robotics and circuit simulators: *CausalCircuit* couples MuJoCo robot arms with digital‑logic models so that perception, action, and outcome are co‑simulated within a single loop. **(iv)** Pushing for visual realism, high‑fidelity 3‑D renderers, like Blender Cycles, underpin *3DIdent*, *Causal3DIdent*, and *CAUSAL3D*, where photorealistic materials, lighting, and multi‑view cameras expand latent spaces from a handful of factors to dozens.
>
> Beyond Blender, *CausalVerse* employs the more powerful simulator engines, such as Unreal Engine 4, to generate high-quality image/video data. For the robotics interaction, the Habitat-Sim and RoboSuite platforms are used for simulation, which are based on the Bullet and MuJoCo physics engines, respectively.
>
> > **Q3**: The paper evaluates a few representative CRL methods. It would have been better to have the authors compare more state-of-the-art methods.
>
> **A3**: Thank you for the suggestion! We fully agree that a broader and more detailed comparison can offer deeper insights into the strengths and limitations of existing methods. In light of your suggestions, we incorporate additional evaluation metrics such as $R^2$, and include new image/video based baseline methods.  These results are summarized in the following tables. We are continuously working to make the benchmark more complete and informative. If there are particular methods or analyses you feel should be included, we would be glad to consider and integrate them.
>
> **Mean Correlation Coefficient (MCC) on Image-based Scenes**
> | Algorithm| Ball on the Slope | Spring | Refraction |
> |-----------|------------------|--------|------------|
> | Supervised| 0.9878| 0.9970 | 0.9900 |
> | iMSDA[1]| 0.4434| 0.6092 | 0.6778 |
> | Sparsity[2] | 0.2491| 0.3353 | 0.1836 |
> | Self-supervised[3]| 0.4109| 0.4523 | 0.3363 |
> | Contrastive Learning[4] | 0.2853|  0.6342 | 0.3773 |
>
> **$R^2$ on Image-based Scenes**
> | Algorithm | Ball on the Slope | Spring | Refraction |
> |-----------|------------------|--------|------------|
> | Supervised|  0.9962| 0.9910 | 0.9800 |
> | iMSDA[1]| 0.9630| 0.9344 | 0.8420 |
> | Sparsity[2]| 0.3242| 0.2340 | 0.4067 |
> | Self-supervised[3]| 0.9658| 0.7841 | 0.7841 |
> | Contrastive Learning[4]|0.9604|0.9920|0.9677 |
>
> **Mean Correlation Coefficient (MCC) on Video-based Scenes**
> | Algorithm | Fall Simple |  Robotics Study |
> | --------- | ----------------- | ------ |
> | IDOL[5] | 0.2527| 0.2500 |
> | CaRiNG[6] | 0.2280| 0.2225 |
> | TDRL[7] | 0.2003 | 0.2440 |
> | TCL[8] | 0.1717 | 0.2163 |
> | iVAE[9]| 0.1881 | 0.1948 |
>
> **$R^2$ on Video-based Scenes**
> | Algorithm | Fall Simple |  Robotics Study |
> | --------- | ----------------- | ------ |
> | IDOL[5]| 0.5901 | 0.6503 |
> | CaRiNG[6]| 0.5457 | 0.6476 |
> | TDRL[7] | 0.5525 | 0.6394 |
> | TCL[8] | 0.4892| 0.6150 |
> | iVAE[9]  | 0.5233| 0.6165 |
>
> [1]Kong L, et al. Partial identifiability for domain adaptation, 2023.
>
> [2]Lachapelle S, et al. Disentanglement via mechanism sparsity regularization: A new principle for nonlinear ICA, 2022.
>
> [3]Von Kügelgen J, et al. Self-supervised learning with data augmentations provably isolates content from style, 2021.
>
> [4] Buchholz S, et al. Learning linear causal representations from interventions under general nonlinear mixing, 2023.
>
> [5] Zijian Li, et al. On the identification of temporally causal representation with instantaneous dependence, 2024.
>
> [6] Guangyi Chen, et al. Caring: Learning temporal causal representation under non-invertible generation process, 2024.
>
> [7] Weiran Yao, et al. Temporally disentangled representation learning, 2022.
>
> [8] Aapo Hyvarinen and Hiroshi Morioka. Unsupervised feature extraction by time-contrastive learning and nonlinear ICA, 2016.
>
> [9] Ilyes Khemakhem, et al. Variational autoencoders and nonlinear ica: A unifying framework. 2020.
>
> > **Q4**:The dataset is limited to computer vision only.
>
> **A4**:  Thank you for the comment. This dataset serves as a starting point, and we chose to showcase images and videos because visual data offers the most direct and effective path toward realism. Even with only image and video data, CausalVerse (as shown in Table 1) already offers richer modality diversity compared to previous datasets that typically include only a single modality. We are enthusiastic about incorporating additional data types, such as text, audio, and time-series, in future updates of CausalVerse to further support comprehensive causal representation learning research.
>
> > **Q5**: I suggest that authors take an example of a dataset and illustrate its different elements, like causal variables and relationships, for visual intuition about the dataset. This can be added to the supplementary material.
>
> **A5**: Thank you for the helpful suggestion. We have added a dedicated section in the appendix illustrating an example scene to showcase its key elements, including causal variables and relationships, for better visual intuition. In our Fall Simple scene, global variables include properties such as the object’s category(e.g., cube, sphere) and acceleration of gravity. Dynamic variables, those that evolve over time, include the position and rotation of the objects, recorded at each simulation timestep. As time progresses, under the influence of gravitational acceleration, the object’s vertical velocity increases, leading to increasingly larger changes in vertical displacement between frames.  The variables for the Fall Simple scene are listed in the table below:
>
>
> | Category | Sub-category | Variable | Dim.  | Type | Range  | Description |
> | :------- | :----------- | :------------- | :---- | :--- | :------------- | :--------------------------------------- |
> | Global | Scene| `scene`| (1,)  | D    | 6 types | Scene name/identifier |
> | Global | Scene| `gravity` | (1,)  | C    | -  | Acceleration of gravity|
> | Global | Object | `render_asset` | (1,)  | D| 90 types | Specifies visual appearance of the object |
> | Dynamic| Object | `position` | (T,3) | C    | - | 3D coordinates across time |
> | Dynamic| Object| `rotation`  | (T,3) | C    | -| Euler angles across time|
>
> Details regarding the data size, causal graph, and description for dynamic physical process analysis can be found in Appendix Table B.3. To further incorporate your valuable suggestion, we will also include a video showcase for this and other scenes in the updated documentation.

---

### Official Review · Reviewer_Ey11 · 2025-06-24

**Rating:** 5
**Confidence:** 5

**Summary:**

This paper presents CausalVerse, a configurable, high-fidelity visual benchmark for causal representation learning (CRL). The dataset spans four domains with 200k images and 300M video frames, each annotated with ground-truth causal structures. It supports both idealized and realistic settings, enabling controlled evaluations under varying assumptions. Empirical comparisons across representative CRL methods reveal notable performance gaps, especially under assumption violations. The benchmark offers a valuable testbed for advancing CRL approaches.

**Dataset Code Accessibility:**

Yes

**Dataset Code Comments:**

The dataset is fully accessible. However, the documentation is too brief.

**Ethical Considerations:**

No, there are no or only very minor ethics concerns

**Final Justification:**

All my concerns are addressed, and I'm maintaining the positive score.

**Limitations Weaknesses:**

1. While there remains a domain gap between Blender-rendered images and real-world scenarios, which may influence the evaluation of latent factor identifiability, the authors have acknowledged this limitation and proposed mitigation strategies in Section 5. Therefore, this issue does not significantly undermine the overall contribution.
2. The documentation in the code repository is overly brief, which may hinder usability and accessibility for potential users of the dataset.

**Strengths Contributions:**

1. This paper presents a large-scale dataset with annotated latent factors tailored for causal representation learning, which is of substantial value for evaluating identifiability in representation learning across various domains.
2. The manuscript is well-structured and clearly written.
3. The code repository is well-organized, and the annotations within the dataset are clear and consistent, making it a meaningful contribution to the community.
4. The appendix provides detailed causal graphs for each scene, along with corresponding latent factors and a thorough description of the experimental setup.

---

> ### Author Rebuttal · Authors · 2025-07-30
>
> Dear Reviewer Ey11
>
> We are grateful for your recognition of our dataset’s contribution and the clarity of the manuscript, and the time you dedicated to reviewing our paper.  Your helpful reminder regarding code documentation has certainly guided us in improving the clarity and usability of our released code. In light of your suggestions, we have added a discussion on the domain gap between Blender-rendered images and real-world scenarios, and we have included documentation to better instruct users on how to use our code. Please see the one-to-one response below:
>
> > **Q1**: While there remains a domain gap between Blender-rendered images and real-world scenarios, which may influence the evaluation of latent factor identifiability, the authors have acknowledged this limitation and proposed mitigation strategies in Section 5. Therefore, this issue does not significantly undermine the overall contribution.
>
>  **A1**: Thank you for highlighting this point and for recognizing our efforts to mitigate the domain gap. To address this gap, we have taken several key steps. First, we incorporate more realistic and complex scenarios. For example, CausalVerse includes settings with 3 to 129 causal variables, an order of magnitude more than previous datasets, which typically feature fewer than 10 variables. Second, we leverage advanced rendering engines such as Blender and Unreal Engine 4, which support industry-leading physically based rendering, real-time ray tracing, and global illumination. These capabilities allow us to generate highly photorealistic images and videos that better approximate real-world visual complexity. While current rendering technologies still have limitations, we are fortunate to be at a time of rapid advancement in simulation, physically based rendering, and AI-generated content. These developments offer new opportunities to push the boundaries of synthetic realism. We view CausalVerse as a foundational step. As the underlying technologies continue to evolve, we are committed to refining and expanding the dataset to provide the research community with a robust platform for developing, testing, and scaling causal representation learning methods in environments that are both controllable and increasingly realistic.
>
> In light of your suggestions, we have added a discussion in Section 5 to further elaborate on the domain gap between Blender-rendered and real-world images. We explicitly acknowledge this limitation to help users better understand the scope and intended use of our dataset.
>
>
> > **Q2**: The documentation in the code repository is overly brief, which may hinder usability and accessibility for potential users of the dataset.
>
> **A2**: Thank you for your kind reminder on the code documentation. We have updated the documentation on our project website to provide clearer and more detailed descriptions of the datasets, making them easier to access and understand for users. Unfortunately, due to the rebuttal policy, we cannot update the repository or revise the paper directly. We promise that all documentation will be released as soon as possible when the rebuttal policy is finished. Here, we show two examples of the  simplified content of some documentation, including one model evaluation script and one dataset metadata structure:
>
> **A2.1**:  Model evaluation script (taking the image-based baseline usage as an example):
>
> **Build Environment**
>
> To install the environment to reproduce the evaluation, please install the package with conda by running `conda env create -f causalverse.yaml`. (Make sure conda is installed properly.)
>
> **Code Structure**
>
> Each baseline method (e.g., CRL_SC, CRL_SP) includes `train.py` (training loop and argument parsing), `basic_model.py` (model definition), and `metrics.py` (evaluation logic).
>
> **Training Scripts**
>
> The script is executed via `python train.py`, with key arguments such as `--meta_csv`, which specifies the metadata CSV containing image paths and ground-truth latent variables.
>
> Taking CRL_SC as an example, the latent vector `z` is divided into view-specific `z_s` (whose dimension is `s_dim`) and view-invariant `z_c` (whose dimension is `z_c = z_dim - s_dim`). In the Cylinder-Spring dataset, `z_c` captures 5 view-invariant causal factors. The training can be launched as follows:
> `python train.py --meta_csv /path/to/Spring.csv --z_dim 15 --s_dim 10`
>
>
>
> **A2.2**:  Dataset Structure (taking the image-based dynamic physical simulations domain as an example)
>
> **Scenes**
>
> The image part of the dynamic physical simulations dataset contains four scenes (Cylinder Spring, Light Refraction, Ball on the Slope, Ball Meets Plasticine), each with 40,000 images rendered from 4 viewpoints. The data folder structure is shown below:
>
> ```
> |–– Static Image Generation/
> ...
> |–– Dynamic Physical Simulations/
> |   |–– Image-based simulation/
> |   |   |–– Light Refraction/
> |   |   |   |–– Light Refraction.csv
> |   |   |   |–– Images
> |   |   |–– Cylinder Spring/
> |   |   |–– Ball on the Slope/
> |   |   |–– Ball Meets Plasticine/
> |   |–– Video-based simulation/
> ...
> |–– Traffic Situation Analysis/
> |–– Robotic Manipulations/
> ```
>
> Each sample is annotated in a CSV file (e.g., `Light Refraction.csv`) with a unique `id`, scene-specific latent variables (e.g., `theta1`, `theta2`, `n1`), a `view` index (0–3), and a `render_path` pointing to the corresponding image.

---

> > ### Comment · Reviewer_Ey11 · 2025-08-05
> > **Response to rebuttal**
> >
> > Thanks for providing the detailed response. All my concerns are addressed. I'm maintaining the positive score.

---

> > > ### Author Response · Authors · 2025-08-05
> > > **Thanks for your feedback!**
> > >
> > > Dear Reviewer Ey11,
> > >
> > > We are delighted that your concerns have been addressed. Thank you again for your recognition and encouragement. We greatly appreciate your thoughtful suggestions to enhance the paper’s readability and the dataset’s usability.
> > >
> > > Best regards,
> > >
> > > The authors

---

### Official Review · Reviewer_de4k · 2025-06-30

**Rating:** 5
**Confidence:** 3

**Summary:**

The paper introduces CausalVerse, a large‐scale, high-fidelity, fully synthetic visual benchmark for Causal Representation Learning (CRL).  The dataset contains ≈ 200 k images and ≈ 140 k videos (≈ 300 M frames) across 24 scenes grouped in four domains: static image generation, dynamic physical simulations, robotic manipulation, and traffic scenarios.  Every scene is generated with Blender, Bullet, Robosuite, or UE4 and is accompanied by ground-truth causal graphs (3–129 latent variables).  Users can access or re-simulate episodes to satisfy or violate common CRL assumptions (e.g., sufficient changes, sparsity, intervention histories).  The authors benchmark five representative CRL methods (CRL_SC, CRL_SP, CRL_SF, CaRiNG, IDOL) plus a supervised upper bound, and discuss when the methods succeed or fail.  Code and data are said to be released on HuggingFace.

**Additional Feedback:**

In my personal view, this is an incremental work; I need more evidence to support the significance of this work.

**Dataset Code Accessibility:**

Yes

**Dataset Code Comments:**

The code is accessible and without issue.

**Ethical Comments:**

No significant ethical concerns remain.

**Ethical Considerations:**

No, there are no or only very minor ethics concerns

**Final Justification:**

The authors have effectively clarified the points that I was concerned about.

**Limitations Weaknesses:**

1. All data are rendered scenes (Sec. 3.3). Domain shift to real images/videos is unquantified, so practical relevance may remain limited.
2. Causal mechanisms are manually designed (lines 142–176), which may encode subjective simplifications. It is unclear how complex or realistic these graphs are, and no quantitative complexity metric is reported.
3. Only three image CRL and two video CRL baselines; Analysis stops at MCC scores—no ablation on assumption violations except sufficient-change vs. sparsity.

**Strengths Contributions:**

1.  Addresses the long-standing realism-vs-ground-truth dilemma in CRL evaluation (lines 32–42).  Offers a single benchmark spanning static–dynamic, low-dimensional–high-dimensional, single/multi-agent settings.
2.   24 heterogeneous scenes with configurable causal graphs;   larger and visually richer than prior CRL datasets (Table 1).  Hierarchical domain–scene–instance design (Fig. 1, Sec. 3.1) enables systematic variation.
3.   Full access to simulation code and parameters allows users to generate custom splits that satisfy or break theoretical assumptions (Sec. 3.4).  This flexibility is valuable for empirical studies of identifiability claims.
4 Provides initial performance numbers for both image and video CRL under matched and mismatched assumptions (Sec. 4, Tables 2–3, Fig. 5).  Demonstrates that state-of-the-art methods still struggle on realistic visuals, underscoring the benchmark’s difficulty.

---

> ### Author Rebuttal · Authors · 2025-07-31
>
> Dear Reviewer de4k,
>
> We sincerely thank you for your thorough review and thoughtful feedback. We are grateful for your recognition of CausalVerse’s contributions in addressing the realism–ground-truth trade-off in CRL, as well as your appreciation of the dataset’s scope, structure, and flexibility for empirical evaluation.
>
> In response to your comments, we have clarified the manually designed nature of the causal graphs and explicitly acknowledged the abstraction they represent. We have also added a discussion on the domain gap between synthetic and real-world data, and noted this as a limitation for future extension. We agree that broader evaluation and more fine-grained ablations would be valuable and leave this as promising future work. Please see our detailed point-by-point responses below.
>
> > **Q1**: All data are rendered scenes (Sec. 3.3). Domain shift to real images/videos is unquantified, so practical relevance may remain limited.
>
>
> **A1**: Thank you for pointing this out. We acknowledge that our data is generated through a rendering engine, which remains a gap between simulated and real-world scenarios.  We have added this into the revised version to help users understand the limitations and assumptions inherent in using the dataset. This reliance on synthetic environments stems from a core challenge in CRL: the need for access to ground-truth latent variables, which are inherently unobservable in real-world data. As such, rendered environments remain a necessary compromise for studying CRL under controlled conditions.
>
> Despite this limitation, our goal is to narrow the gap by leveraging a more powerful rendering engine capable of producing high-fidelity visuals and by modeling increasingly complex scenarios involving multiple causal variables. While full realism cannot yet be achieved, we are committed to moving in that direction by providing the research community with datasets that are as realistic, diverse, and large-scale as possible to support continued progress in CRL.
>
> > **Q2**: Causal mechanisms are manually designed (lines 142–176), which may encode subjective simplifications. It is unclear how complex or realistic these graphs are, and no quantitative complexity metric is reported.
>
>
> **A2**: We appreciate this insightful question, which allows us to clarify the complexity and realism of our manually designed causal graphs.
> In terms of complexity, the number of causal variables may serve as an effective quantitative proxy. As shown in Table 1, our dataset contains graphs with over 100 variables, which is an order of magnitude more complex than datasets like CAUSAL3D, which typically contain fewer than 10 variables.
> Regarding realism, we acknowledge that there is no universally accepted quantitative metric to evaluate how realistic a causal graph is. However, we designed most of our graphs based on well-established physical constraints (e.g., physical processes, robotic simulations) or social rules (e.g., traffic environments), ensuring that they reflect plausible and interpretable structures within those domains.
>
> > **Q3**: Only three image CRL and two video CRL baselines; Analysis stops at MCC scores—no ablation on assumption violations except sufficient-change vs. sparsity.
>
> **A3**: Thank you for the thoughtful suggestion. We agree that a more extensive comparison can offer deeper insights into model performance. In light of your suggestions, we have added new evaluation metrics such as $R^2$, and included additional (4) image-based and (5) video-based CRL methods. Besides, we have also added the ablation study on assumption violations about the domain request in assumptions (reducing the domain number, or giving wrong domain information). All results have been included in the appendix and copied into the tables below.
>
> We are actively working to make the benchmark more systematic and comprehensive, and we will continue to update our public website with new results and analyses. Thank you again for the helpful reminder. If there are any specific methods or analyses you believe should be added, we would be more than happy to incorporate them.
>
> **Mean Correlation Coefficient (MCC) on Image-based Scenes**
> | Algorithm| Ball on the Slope | Spring | Refraction |
> |-----------|------------------|--------|------------|
> | Supervised| 0.9878| 0.9970 | 0.9900 |
> | iMSDA[1]| 0.4434| 0.6092 | 0.6778 |
> | Sparsity[2] | 0.2491| 0.3353 | 0.1836 |
> | Self-supervised[3]| 0.4109| 0.4523 | 0.3363 |
> | Contrastive Learning[4] | 0.2853|  0.6342 | 0.3773 |
>
>
> **$R^2$ on Image-based Scenes**
> | Algorithm | Ball on the Slope | Spring | Refraction |
> |-----------|------------------|--------|------------|
> | Supervised|  0.9962| 0.9910 | 0.9800 |
> | iMSDA[1]| 0.9630| 0.9344 | 0.8420 |
> | Sparsity[2]| 0.3242| 0.2340 | 0.4067 |
> | Self-supervised[3]| 0.9658| 0.7841 | 0.7841 |
> | Contrastive Learning[4]|0.9604|0.9920|0.9677 |
>
>
> **Mean Correlation Coefficient (MCC) on Video-based Scenes**
> | Algorithm | Fall Simple |  Robotics Study |
> | --------- | ----------------- | ------ |
> | IDOL[5] | 0.2527| 0.2500 |
> | CaRiNG[6] | 0.2280| 0.2225 |
> | TDRL[7] | 0.2003 | 0.2440 |
> | TCL[8] | 0.1717 | 0.2163 |
> | iVAE[9]      | 0.1881            | 0.1948 |
>
> **$R^2$ on Video-based Scenes**
> | Algorithm | Fall Simple |  Robotics Study |
> | --------- | ----------------- | ------ |
> | IDOL[5]| 0.5901 | 0.6503 |
> | CaRiNG[6]| 0.5457 | 0.6476 |
> | TDRL[7] | 0.5525 | 0.6394 |
> | TCL[8] | 0.4892| 0.6150 |
> | iVAE[9]      | 0.5233            | 0.6165 |
>
> **Ablation on Assumption Violations in Static Image Generation Domain**
> | Algorithm | Four-Scenes (MCC/R²) | One-Scene (MCC/R²) | Wrong Scene Labels (MCC/R²) |
> |-----------|----------------------|---------------------|------------------------------|
> | SUP       | 0.9001 / 0.6882      | 0.8646 / 0.5808     | 0.9001 / 0.6882              |
> | iMSDA[1]    | 0.3264 / 0.2898      | 0.3197 / 0.2671     | 0.1412 / −6.7706             |
> | Sparsity[2]    | 0.1423 / 0.1681      | 0.1538 / 0.2035     | – / –                        |
>
>
> [1]Kong L, et al. Partial identifiability for domain adaptation, 2023.
>
> [2]Lachapelle S, et al. Disentanglement via mechanism sparsity regularization: A new principle for nonlinear ICA, 2022.
>
> [3]Von Kügelgen J, et al. Self-supervised learning with data augmentations provably isolates content from style, 2021.
>
> [4] Buchholz S, et al. Learning linear causal representations from interventions under general nonlinear mixing, 2023.
>
> [5] Zijian Li, et al. On the identification of temporally causal representation with instantaneous dependence, 2024.
>
> [6] Guangyi Chen, et al. Caring: Learning temporal causal representation under non-invertible generation process, 2024.
>
> [7] Weiran Yao, et al. Temporally disentangled representation learning, 2022.
>
> [8] Aapo Hyvarinen and Hiroshi Morioka. Unsupervised feature extraction by time-contrastive learning and nonlinear ICA, 2016.
>
> [9] Ilyes Khemakhem, et al. Variational autoencoders and nonlinear ica: A unifying framework. 2020.

---

### Official Review · Reviewer_C9gi · 2025-07-02

**Rating:** 5
**Confidence:** 3

**Summary:**

The paper focuses on the evaluation problem of Causal Representation Learning (CRL), pointing out that current evaluation methods either rely on overly simplified synthetic datasets or on performance in real-world downstream tasks, leading to a dilemma between realism and evaluation accuracy. This is a very critical and challenging issue. The authors construct a synthetic dataset from four aspects, which can precisely test the performance of models under specific causal assumptions. Experiments show that the proposed dataset is sufficiently challenging and provides a comprehensive testing platform for future researchers.

**Additional Feedback:**

I hope the authors can respond to the points I raised in the Limitations Weaknesses section.

**Dataset Code Accessibility:**

Yes

**Ethical Considerations:**

No, there are no or only very minor ethics concerns

**Final Justification:**

The author has completely resolved my concerns, and I have raised my score to Accept.

**Limitations Weaknesses:**

1. The authors mention that existing methods [33, 5, 34, 18] (as cited in the main paper) design some simple synthetic environments that lack realism. However, the proposed approach in this paper is also based on synthetic data. How do the authors ensure the realism of the datasets they propose?

2. Real-world datasets often contain various types of noise. For example:
   a. The collected visual data may be affected by sensor noise or interference. Has this factor been considered in the construction of the proposed dataset? I believe the current visual data in the dataset might be too "clean" to realistically simulate real-world scenarios.
   b. In the real world, due to the influence of other factors, many situations do not strictly follow the pre-defined causal relationships (e.g., in physical experiments, uncertainties in initial velocity, complexities in material properties, and environmental factors can all interfere with the results).
    I suggest the authors discuss these two points to enhance the robustness of the dataset.

3. The authors state that CausalVerse supports flexible configuration of underlying causal graphs (such as domain labels, temporal dependencies, and intervention histories), but this flexibility does not seem to be demonstrated in the appendix.

4. I am curious about the rationality of the manually constructed causal relationships. How do the authors ensure the reasonableness of such causal structures? For example, how was the causal graph in Table 2 of the appendix designed?

**Strengths Contributions:**

1. The authors propose constructing a high-fidelity simulated visual data benchmark (CausalVerse) to address the aforementioned evaluation dilemma, and this motivation is reasonable. By using simulated data, it is possible to ensure data complexity while providing a known causal generative process, thereby enabling precise evaluation of causal representation learning methods.

2. Moreover, the authors take into account the differing requirements for causal representation learning methods across various application scenarios, and design a dataset that includes multiple scenarios (such as static image generation, dynamic physical simulation, robotic manipulation, traffic analysis, etc.). This allows the benchmark to cover a wide range of causal learning tasks, providing researchers with a comprehensive testing platform.

3. The authors present numerous examples of the datasets they constructed in the appendix, demonstrating substantial effort, and the datasets are sufficiently challenging.

---

> ### Author Rebuttal · Authors · 2025-07-30
>
> Dear Reviewer C9gi,
>
> We sincerely appreciate your encouraging comments on our contributions, your valuable suggestions for enhancing the dataset’s utility, and the time and effort you dedicated to reviewing our paper. In response to your suggestions, we added more discussions about the distinctions, limitations, and explanations. The one-to-one responses can be found below:
>
> > **Q1**: The authors mention that existing methods [33, 5, 34, 18] (as cited in the main paper) design some simple synthetic environments that lack realism. However, the proposed approach in this paper is also based on synthetic data. How do the authors ensure the realism of the datasets they propose?
>
> **A1**: Thanks for your excellent question, as it helped us to clarify the contribution over existing datasets. In light of your suggestions, we have added the following discussion in the revised version to clarify the distinctions.
>
> Different from existing methods [33, 5, 34, 18] which also leverage the synthetic data to simulate the ground-truth causal relations, we highlight two efforts of our CausalVerse towards realism. First, we consider the more realistic and complex scenarios. As a comparison, CausalVerse includes scenarios with more than 100 causal variables (3-129), while previous work has fewer than 10 variables. Second, we apply more powerful rendering engines such as Blender and Unreal Engine 4 to offer industry‑leading physically based rendering to produce photorealistic images and videos, and also use the Habitat-Sim and RoboSuite platforms for robotic manipulation simulation.
>
> > **Q2**:. Real-world datasets often contain various types of noise. For example:
>
>  >   a. The collected visual data may be affected by sensor noise or interference. Has this factor been considered in the construction of the proposed dataset? I believe the current visual data in the dataset might be too "clean" to realistically simulate real-world scenarios.
>
>  >   b. In the real world, due to the influence of other factors, many situations do not strictly follow the pre-defined causal relationships (e.g., in physical experiments, uncertainties in initial velocity, complexities in material properties, and environmental factors can all interfere with the results).
>
>  >   I suggest the authors discuss these two points to enhance the robustness of the dataset.
>
> **A2**: We highly appreciate these insightful questions, which helped us clarify the limitations and assumptions inherent in using the dataset.
>
>    (a) Regarding sensor noise or interference, we have made efforts to account for this during dataset construction. For instance, in our video-based physical simulation scenes, we utilize the Habitat-Sim simulator, which includes a library of noise models specifically designed to bridge the gap between simulated and real sensor observations. This allows us to capture subtle variations in object motion under identical variable settings. While we simulate noise through the rendering engine, we acknowledge that real-world noise is often far more complex, and reproducing it accurately remains a non-trivial challenge. As discussed in the limitations section (Sec. 5), we highlight this potential gap to remind readers and dataset users of the risks associated with relying solely on synthetic data. Fortunately, rendering technologies continue to advance rapidly, offering the opportunity to progressively narrow this gap and bring synthetic data closer to real-world visual conditions.
>
>    (b) We completely agree that in real-world scenarios, causal relationships may not strictly hold due to factors such as uncertainties in initial conditions, variations in material properties, and environmental noise. However, we believe that while these factors may affect the values of variables or the parameters of the functional relationships, the underlying causal directions generally remain valid. For example, even though air resistance alters the motion of a falling object, the object’s acceleration is still fundamentally governed by gravity—albeit in a non-linear fashion. To better reflect such complexities, during the data construction, we have modeled variations in material properties by sampling spheres with different surface roughness to simulate different materials. In light of your suggestion, we have added a discussion in Section 5 explicitly acknowledging that real-world situations may not strictly follow pre-defined causal relationships. We hope this addition helps future researchers better understand the limitations and assumptions inherent in using this dataset.
>
> > **Q3**:. The authors state that CausalVerse supports flexible configuration of underlying causal graphs (such as domain labels, temporal dependencies, and intervention histories), but this flexibility does not seem to be demonstrated in the appendix.
>
> **A3**: Thanks so much! We totally agree that more concrete demonstrations would be helpful to support the flexible configuration. We have added a detailed demonstration in the appendix. Due to the rebuttal policy, we cannot update the repository or revise the appendix to provide a detailed demonstration during the rebuttal. Instead, here, we briefly show some flexible configurations:
>
> * Domain labels: In the static-image part, setting `domain_id = k` automatically swaps the background mesh and HDR lighting, yielding distinct domains. In the free-fall scene, varying the gravitational constant $g ∈ \\{ 4.9, 9.8, 14.7 \\} ~ms^{-2} $ creates domains that differ only in the coefficient of the *height → velocity* edge.
>
> * Temporal dependencies: For traffic videos, toggling `traffic rules such as distance between cars` changes the conditional distribution  `P(Actionₜ | Stateₜ₋₁)`. For physics environments, increasing the physics record time-step from 0.02 s to 0.10 s introduces higher-order dynamics.
>
> * Intervention histories: In robotics, researchers can directly program the robotics to perform a predefined action, e.g., sweeping a full arc around the workspace, thereby intervening on both actions and state trajectories. The intervention call is logged, enabling deterministic replay for counterfactual evaluation. In physics scenes, such as the gravitational constant, spring constant, or restitution coefficient, can be overwritten at arbitrary frames to create *soft* or *hard* interventions.
>
> These configurations enable targeted experiments, e.g., treating different backgrounds as domains when evaluating domain-needed methods while preserving the latent causal graph.
>
> > **Q4**: I am curious about the rationality of the manually constructed causal relationships. How do the authors ensure the reasonableness of such causal structures? For example, how was the causal graph in Table 2 of the appendix designed?
>
> **A4**: We are grateful for this constructive question, which gave us an opportunity to clarify the rationale behind our manually constructed causal relationships. These relationships primarily stem from three sources: (1) physical constraints, (2) basic social rules, and (3) arbitrary human constructs.
>
> For instance, in physics-based processes and robotic domains, the causal dependencies are constructed to align with established physical laws. In the traffic domain, the causal influences reflect social conventions such as avoiding collisions and stopping at red lights. For the static image generation domain (see Table 2 in the appendix), the causal structures are from arbitrary human constructs, which are intentionally designed to allow flexible data manipulation and make relations complex, rather than to represent realistic causality. We acknowledge that these arbitrarily designed causal graphs (only in the static image generation domain) do not reflect real-world causal structures. In these cases, the graphs primarily serve as tools for structured causal relations rather than realistic modeling. In light of your suggestion, we have explicitly clarified this point in the revised appendix, to help users better understand the design goals and intended usage of these graphs.
> Besides, for causal structures derived from physical laws or social rules, we carefully ensure their plausibility and internal consistency during the design process.

---

### Note · Authors · 2025-08-12

Dear AC and Reviewers,

We thank the AC for their time and contributions to the community, and we are grateful to the reviewers for their kind recognition of our work and their invaluable comments and suggestions. In light of your insightful suggestions, we have incorporated all results and discussions into the final version, which we briefly summarize below.

**To Reviewer C9gi (Q1, Q2), Reviewer Ey11 (Q1), and Reviewer de4k (Q1)**, we have acknowledged the gap between the current synthetic data and real-world realism (e.g., the greater complexity of real-world noise) in Section 5 and have clarified our ongoing efforts toward generating more realistic data compared with other existing datasets in Section 1.

**To Reviewer C9gi (Q3)**, we have added concrete configuration examples in Appendix B and will update the repository immediately after the rebuttal period.

**To Reviewer C9gi (Q4)**, we have provided explanations to justify the rationale behind the manually constructed causal relationships in Appendix B.

**To Reviewer de4k (Q2)**, we have reported the complexity of causal graphs in Table 1 and discussed the realism in Section 5.

**To Reviewer de4k (Q3)** and **Reviewer EXTi (Q3)**, we have broadened the evaluation and included them in the final version: added R², 4 image and 5 video baselines (in Tables 2 and 3), and ablations on domain-assumption violations (adding Table 9 in the Appendix).

**To Reviewer Ey11 (Q2)**, we have expanded the project page (env setup, training/eval usage, dataset schema/examples) and will update the repository immediately after the rebuttal.

**To Reviewer EXTi (Q1)**, we have prepared data-generation code and documentation; a brief pipeline is added in Appendix D.

**To Reviewer EXTi (Q2)**, we have added a comparative discussion of simulation approaches versus existing benchmarks in Appendix A.

**To Reviewer EXTi (Q4)**, we have clarified the vision-only scope and left the multi-modal extension as future work.

**To Reviewer EXTi (Q5)**, we have added an illustrative scene example detailing causal variables/relations in Appendix B.

Thanks again to the reviewers and AC for the time dedicated to reviewing this paper.

Sincerely,

The Authors

---

### Decision · Program_Chairs · 2025-09-18

**Decision:**

Accept (spotlight)

**Comment:**

The paper presents a large-scale dataset of well annotated latent factors in static generation, physical simulation, robotic manipulation, and traffic analysis to benchmark the hard problem of Causal Representation Learning (CRL). The scale and the process of creating the dataset are impressive and add a lot of value to the community and help evaluate techniques fairly. While the modality is limited, it is a good step in the right direction and will pave way for better algorithms along the way.

Overall, I recommend accepting the paper, so do all the reviewers. Great work!